# Unsupervised representation learning on high-dimensional clinical data improves genomic discovery and prediction

Taedong Yun [1] ✉, Justin Cosentino [2], Babak Behsaz[1], Zachary R. McCaw [2,12], Davin Hill [3,4], Robert Luben [5,6], Dongbing Lai [7], John Bates[8], Howard Yang[2], Tae-Hwi Schwantes-An[7,9], Yuchen Zhou[1], Anthony P. Khawaja [5,6], Andrew Carroll [2], Brian D. Hobbs [4,10,11], Michael H. Cho [4,10,11], Cory Y. McLean [1,13] ✉ & Farhad Hormozdiari [1,13] ✉

Although high-dimensional clinical data (HDCD) are increasingly available in biobank-scale datasets, their use for genetic discovery remains challenging. Here we introduce an unsupervised deep learning model, Representation Learning for Genetic Discovery on Low-Dimensional Embeddings (REGLE), for discovering associations between genetic variants and HDCD. REGLE leverages variational autoencoders to compute nonlinear disentangled embeddings of HDCD, which become the inputs to genome-wide association studies (GWAS). REGLE can uncover features not captured by existing expert-defined features and enables the creation of accurate disease-specific polygenic risk scores (PRSs) in datasets with very few labeled data. We apply REGLE to perform GWAS on respiratory and circulatory HDCD—spirograms measuring lung function and photoplethysmograms measuring blood volume changes. REGLE replicates known loci while identifying others not previously detected. REGLE are predictive of overall survival, and PRSs constructed from REGLE loci improve disease prediction across multiple biobanks. Overall, REGLE contain clinically relevant information beyond that captured by existing expert-defined features, leading to improved genetic discovery and disease prediction.

Modern healthcare systems generate a vast amount of high-dimensional clinical data (HDCD), such as spirograms, photoplethysmogram (PPG), electrocardiogram (ECG), computed tomography and magnetic resonance imaging, that cannot be summarized as a single binary or a continuous number (such as 'has asthma' or 'height in centimeters'). HDCD provide a unique opportunity to reveal the genetic architecture of diseases and complex traits when coupled with biobank-scale genetic data[1–6], but we lack statistical methods to fully use HDCD in genome-wide association studies (GWAS), as standard GWAS require the phenotype of interest to be encoded as a single scalar.

The most common method for GWAS on HDCD uses a small number of expert-defined features (EDFs) extracted from the HDCD as the target phenotypes. For example, spirograms are a graphical representation of spirometry test results, a widely used clinical test for lung function that measures airflow and volume over time[7,8]. Spirograms can be summarized into EDFs, including forced vital capacity (FVC), forced expiratory volume in the first second ($FEV_1$), $FEV_1$/FVC (nonlinear function of FVC and $FEV_1$), peak expiratory flow (PEF) and forced mid-expiratory flow ($FEF_{25–75\%}$)[9]. Spirogram EDFs are used in clinical settings to diagnose diseases such as chronic obstructive pulmonary disease (COPD)[10,11]. In another example, PPG measures volumetric

**Fig. 1 | Overview of REGLE.** In step 1, we learn a low-dimensional embedding using a VAE, where we optionally condition the decoder on EDFs. In step 2, we perform GWAS on all learned coordinates (and EDFs if they are used). Finally, in step 3, we train a small linear model to learn weights for each latent coordinate PRS to obtain the final disease-specific PRS.

changes in peripheral blood circulation using infrared light. Previously studied EDFs of PPG include the presence (or absence) of a notch, position of the notch, position of the peak, position of the shoulder and peak-to-peak time[12–16]. PPG EDFs have known associations with cardiovascular diseases, such as coronary heart disease[12]. Spirograms and PPG EDFs are heritable, and GWAS on EDFs have helped identify the genetic architecture of lung[17–19] and circulatory function[20–22]. However, EDFs may not capture all heritable signals encoded in spirograms or PPGs, thus GWAS on these EDFs may not exploit the full potential of these HDCD.

A simple approach to HDCD GWAS performs GWAS on each data coordinate (for example, time point or pixel). For example, previous work performed GWAS on each recorded ECG time point[23]. This approach is computationally expensive and has low statistical power due to the high correlation of nearby coordinates and the massive multiple-testing burden[24,25]. A popular alternative performs principal component analysis (PCA)[26] on the HDCD and then GWAS on a subset of the PCs[27]. However, PCA assumes a linear relationship between the raw HDCD and the underlying biological factors of interest and does not explicitly model spatial or temporal structure. Moreover, performing GWAS on a subset of PCs may miss heritable signals, which are often small.

Machine learning (ML)-based phenotyping uses HDCD as input to a supervised ML model to predict trait labels and then performs GWAS using the model predictions as the target phenotype[3,6,28]. While ML-based phenotyping can augment standard GWAS on manually defined trait labels, the supervised model only learns signals related to the specific target trait. Additionally, for the common case in which the supervised model uses deep learning, many labeled examples may be required to achieve good performance.

To overcome these limitations, we developed a principled method, Representation Learning for Genetic Discovery on Low-Dimensional Embeddings (REGLE), that is computationally efficient, requires no labels and can incorporate information from EDFs if available. REGLE is based on the variational autoencoder (VAE)[29] model. Although VAEs

have previously been applied to metabolomics data[30], the utility of VAE embeddings for GWAS, polygenic risk scores (PRSs) and downstream analyses has not been previously explored. We apply REGLE in two case studies to understand the genetic architecture of lung function from raw spirograms and circulatory function from PPG. Compared to GWAS on spirogram and PPG EDFs, our GWAS on the learned encodings recovers the most known genetic loci linked to lung and circulatory function while also detecting additional loci. PRS created from loci identified via GWAS of REGLE of spirograms improves COPD and asthma predictions. Similarly, PRSs derived from REGLE of PPG improve hypertension (HTN) and systolic blood pressure (SBP) predictions. These results indicate that REGLE successfully extracts a meaningful representation of lung function from spirograms and of circulatory function from PPG, which in turn improves genetic discovery and risk prediction.

## Results

### Overview of REGLE

REGLE consists of three main steps. First, we learn a nonlinear, low-dimensional, disentangled representation (that is, an encoding) of the HDCD using a VAE[29] trained to compress and reconstruct HDCD (Fig. 1; Methods). Autoencoders consist of an encoder and a decoder, connected by a low-dimensional 'bottleneck' layer. The encoder summarizes the input data into a small set of numbers at the bottleneck layer, and the decoder reconstructs the input data from the low-dimensional summary[31]. VAE[29] is a special type of autoencoder that introduces stochasticity in the encoder. The VAE implicitly forces the learned encodings to be relatively disentangled[32], that is, the encodings have relatively uncorrelated coordinates and separable biological factors can be better captured in each coordinate. Second, we perform GWAS independently on each encoding coordinate. Third, we use PRSs from the encoding coordinates as genetic scores of general biological functions and potentially combine them to create a PRS for a disease or trait of interest (Fig. 1).

REGLE enables relevant EDFs to be optionally included in the input to the decoder of the model so that the encoder is encouraged to learn

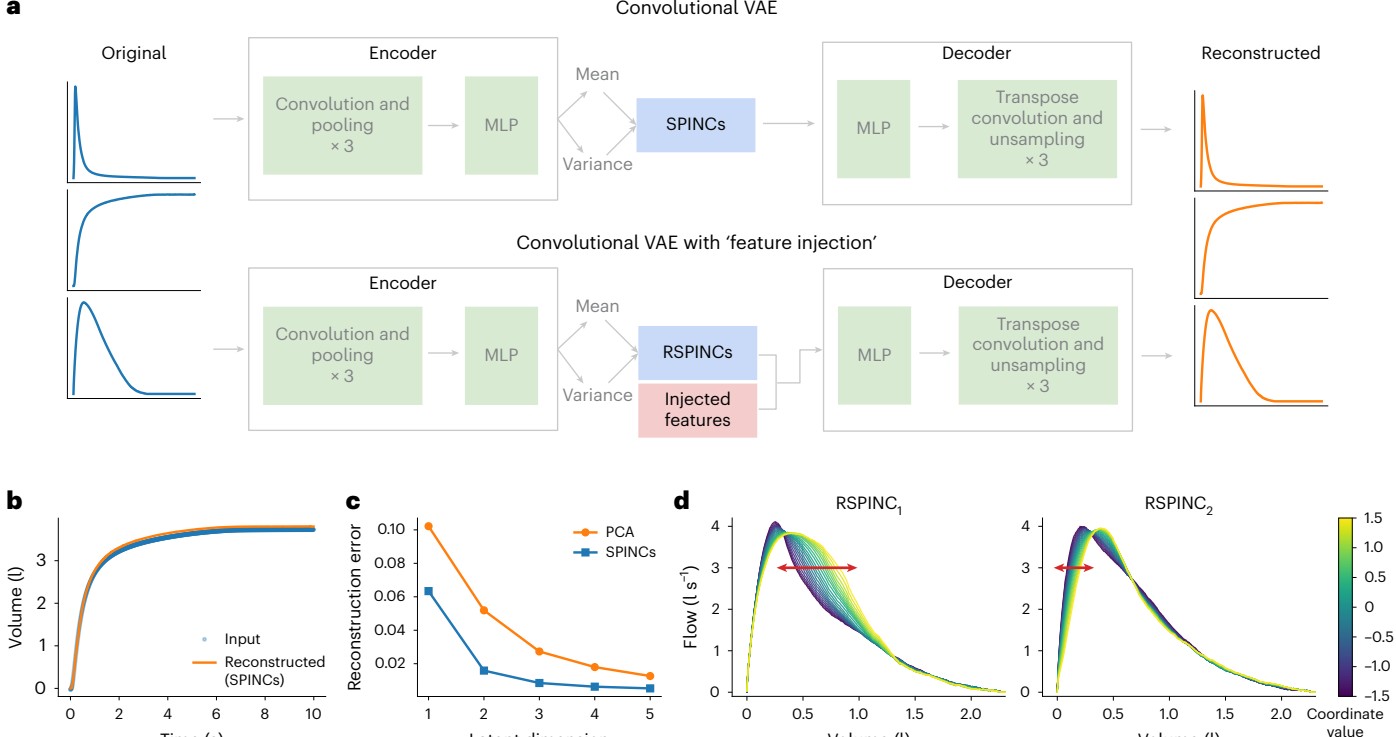

**Fig. 2 | Overview of REGLE on spirograms. a**, Learning SPINCs using a convolutional VAE and RSPINCs using a convolutional VAE with 'feature injection', for example, using EDFs. **b**, Reconstructing a spirogram (volume–time curve) from SPINCs (dim = 5). **c**, Reconstruction errors (mean squared error across time points) for reconstructed spirograms using the SPINCs model (blue) and PCA (orange) with a varying latent dimension. Both the SPINCs model and PCA are trained (or 'fitted') on a training set, and the reconstruction error is evaluated in a separate validation set. **d**, Spirograms created by RSPINCs (dim = 2) decoder using a fixed set of injected features (that is, EDFs) and varying one RSPINC coordinate while fixing the other one to be zero. Line color indicates the varying RSPINC coordinate value from low (blue) to high (yellow).

only the residual signals not represented by the EDFs (Fig. 1). This ability to incorporate prior knowledge of important data features (from users or clinicians) is a key advantage of REGLE.

## Overview of REGLE on spirograms

We applied REGLE to obtain low-dimensional representations of spirogram curves, which we call spirogram encodings (SPINCs; Fig. 2). To construct SPINCs, we trained a convolutional VAE[29] to reconstruct spirograms (Fig. 2a; Methods). In addition, we constructed another set of encodings we call residual spirogram encodings (RSPINCs) by injecting five EDFs (FEV$_1$, FVC, FEV$_1$/FVC, PEF and FEF$_{25-75\%}$) as inputs to the decoder when reconstructing flow–volume curves (Fig. 2a). We generated SPINCs and RSPINCs for all individuals ($n = 351,120$) in the UK Biobank (UKB)[33,34] using their first-visit spirogram, excluding individuals whose spirogram failed our quality control (QC) measures (Methods). We used 80% of the individuals whose genetically inferred ancestry (GIA) is European ($n = 259,692$) to train the (R)SPINCs models and 20% ($n = 65,266$) to evaluate reconstruction performance and choose hyperparameters (Extended Data Fig. 1 and Supplementary Table 1; Methods). Using just five SPINCs (the number of common spirogram EDFs), we observed highly accurate reconstruction of the input spirograms (Fig. 2b). SPINCs consistently outperformed an equivalent number of PCs in terms of reconstruction accuracy at small latent dimensions (Fig. 2c, Supplementary Table 2 and Supplementary Note). We observed similarly accurate reconstructions using EDFs + RSPINCs and confirmed that the addition of RSPINCs improves the reconstruction quality significantly, compared to using a decoder-only model to reconstruct curves from EDFs only (Extended Data Fig. 2). We used two RSPINCs to balance the number of additional coordinates and the reconstruction accuracy. Notably, the learned representations are

highly consistent when trained with multiple different initializations (Extended Data Fig. 3 and Supplementary Note).

## (R)SPINCs are partially interpretable

Leveraging the generative nature of REGLE models, we studied the influence of RSPINC coordinates on spirogram shape by fixing the values of EDFs (obtained from a randomly selected individual in the validation set) and varying one RSPINC coordinate while keeping the other one fixed at zero and generating the corresponding flow–volume spirograms using only the decoder portion of the RSPINCs model (Fig. 2d). A typical flow–volume spirogram consists of the following two distinct parts: a relatively brief part to reach peak flow where the flow increases monotonically as the volume increases, and the main part of the spirogram where the flow decreases monotonically. In Fig. 2d, we clearly observed that varying the first coordinate of RSPINCs amounts to widening or narrowing of the second part (negative slope) while keeping the first part relatively fixed. Similarly, varying the second coordinate of RSPINCs widens or narrows the first part (positive slope) while keeping the second part relatively fixed. Notably, when varying either coordinate, the maximum flow value (PEF) and the final volume value (FVC) stay roughly the same, as expected because all EDFs were fixed.

## Overview of REGLE on PPGs

We applied REGLE to obtain low-dimensional representations of PPG curves computed from a median single heartbeat, which we call PPG encodings (PLENCs; Fig. 3). To construct PLENCs, we trained a convolutional VAE[29] to reconstruct PPGs (Fig. 3a; Methods). We generated PLENCs for all individuals ($n = 170,714$) in UKB[33] using their first-visit PPG, excluding individuals whose PPG failed our QC measures (Methods). We used 80% of the European GIA individuals ($n = 136,239$)

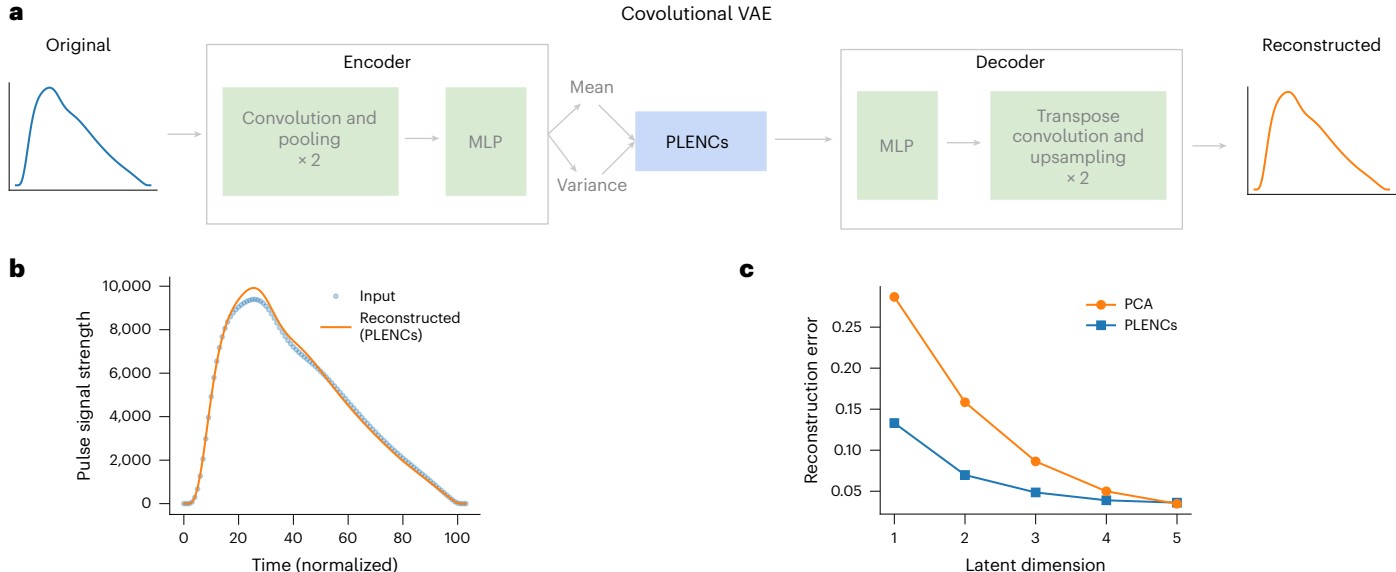

**Fig. 3 | Overview of REGLE on PPG. a**, Learning PLENCs using a convolutional VAE. **b**, Reconstructing PPG from PLENCs (dim = 5). **c**, Reconstruction errors (mean squared error across time points) for reconstructed PPGs using the PLENCs model (blue) and PCA (orange) with a varying latent dimension. Both the PLENCs model and PCA are trained (or 'fitted') on a training set, and the reconstruction error is evaluated in a separate validation set.

**Table 1 | Comparison of GWAS significant loci**

| System | Method (number of traits) | Sample size | Total | Known (%) | Unknown (%) |
|---|---|---|---|---|---|
| Lung | GWAS Catalog+Shrine et al.[19] | – | 1,104 | – | – |
| | Shrine et al.[19] | 581×10³ | 754 | – | – |
| | Spirogram EDFs (5) | 325×10³ | 613 | 581 (95%) | 32 (5%) |
| | Spirogram PCA (5) | 325×10³ | 412 | 397 (96%) | 15 (4%) |
| | SPINCs (5) | 325×10³ | 575 | 510 (89%) | 65 (11%) |
| | EDFs+RSPINCs (7) | 325×10³ | 659 | 596 (90%) | 63 (10%) |
| Cardiovascular | GWAS Catalog | – | 520 | – | – |
| | PPG EDFs (5) | 141×10³ | 62 | 24 (39%) | 38 (61%) |
| | PPG PCA (5) | 141×10³ | 43 | 20 (47%) | 23 (53%) |
| | PLENCs (5) | 141×10³ | 90 | 40 (44%) | 50 (56%) |
| | EDFs+RPLENCs (7) | 141×10³ | 75 | 28 (37%) | 47 (63%) |

For lung function and spirograms, EDFs are $FEV_1$, FVC, $FEV_1$/FVC, PEF and $FEF_{25-75\%}$, and 'known' and 'unknown' are in reference to lung function loci in the GWAS Catalog and Shrine et al.[19]. For cardiovascular function and PPG, EDFs are absence of notch, position of notch, position of peak, position of shoulder and peak-to-peak time, and 'known' and 'unknown' are in reference to cardiovascular disease loci in GWAS Catalog.

to train the PLENCs models and 20% ($n$ = 34,475) to evaluate the reconstruction performance and choose hyperparameters (Extended Data Fig. 4 and Supplementary Table 1; Methods). With just five PLENCs (the number of PPG EDFs), we observed a highly accurate reconstruction of the input PPG (Fig. 3b and Supplementary Table 3). PLENCs consistently outperformed PCs in terms of reconstruction accuracy at small latent dimensions (Fig. 3c and Supplementary Note). We also constructed residual PPG encodings (RPLENCs) by injecting five PPG EDFs (absence of notch, position of notch, position of peak, position of shoulder and peak-to-peak time).

## (R)SPINCs and (R)PLENCs encode information beyond EDFs
Some SPINCs and PLENCs are highly correlated with known EDFs (Pearson correlation $r$ between $SPINC_3$ and FVC is 0.96; $r$ between $PLENC_3$ and position of the shoulder is 0.74; Extended Data Figs. 5 and 6), while both RSPINCs coordinates have low correlation ($|r| < 0.3$) with EDFs as expected (Extended Data Fig. 5). (R)SPINCs and (R)PLENCs are also correlated with other predictors of lung function (covariates), such as age, sex, height, body mass index and smoking status (Extended Data Fig. 5).

We residualized both the EDFs and the covariates from (R)SPINCs and (R)PLENCs and computed correlation with tabular UKB features (UKB phenotypes whose types are a real number, integer, date, binary or categorical). Multiple groups of fields strongly and significantly correlated with the (R)SPINCS and (R)PLENCs even after residualizing (Supplementary Tables 4–8 and Supplementary Note). Both (R)SPINCs and (R)PLENCs were associated with overall survival. For example, $SPINC_3$ had a hazard ratio of 0.68 (95% confidence interval (CI), 0.65–0.71; $P = 1.6 \times 10^{-83}$ under the Cox proportional hazards model), implying the hazard of death decreased by 32% per one s.d. increase in the coordinate (Supplementary Note, Extended Data Fig. 7, Supplementary Figs. 1–3 and Supplementary Table 9; Methods).

## REGLE detects new loci for lung and circulatory functions
We generated SPINCs (dim = 5), RSPINCs (dim = 2, in addition to five EDFs) for all individuals with valid first-visit spirograms in UKB (Extended Data Fig. 1 and Supplementary Figs. 4 and 5; Methods) and PLENCs (dim = 5), RPLENCs (dim =2, in addition to five EDFs) for all individuals with valid first-visit PPGs in UKB (Extended Data Fig. 4;

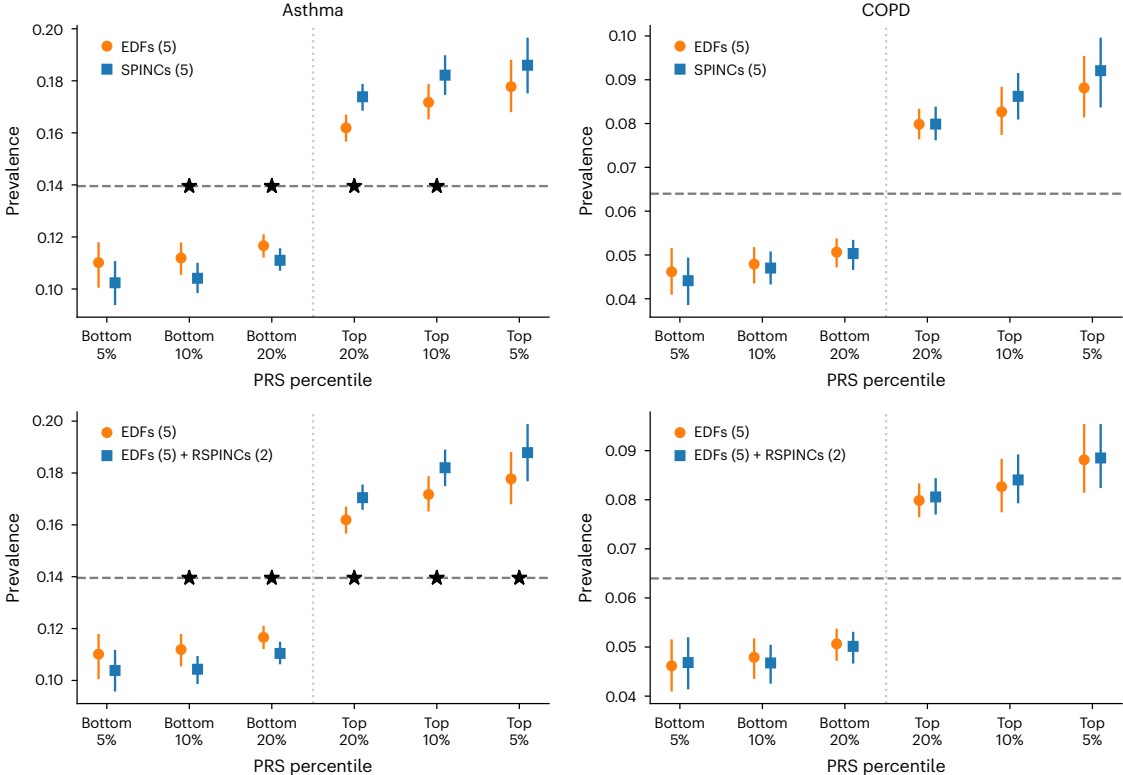

**Fig. 4 | PRS using SPINCs and RSPINCs in UKB.** Combined PRS for medical-record-based asthma and COPD using the following three sets of intermediate PRS: five EDFs, five SPINCs and five EDFs + two RSPINCs. Each set of PRS is combined by a linear model trained using the target phenotype labels, and the prevalence of the phenotypes in the top and bottom 5%, 10% and 20% PRS individuals is evaluated in a separate evaluation set. Vertical line segments indicate 95% CIs generated by bootstrapping (300 repetitions), and the center points are the bootstrapping means. The horizontal dashed lines show the total prevalence. Star symbols indicate a statistically significant difference between the two methods using paired bootstrapping (300 repetitions) with 95% confidence (that is, two-sided $P < 0.05$). Lower is better for the bottom percentiles; higher is better for the top percentiles.

Methods). We then performed GWAS on all European GIA individuals across all encoding coordinates, five spirogram EDFs and five PPG EDFs using BOLT-LMM[35,36], adjusting for covariates (Supplementary Note and Supplementary Figs. 6–19; Methods). (R)SPINCs and (R)PLENCs have significant SNP heritability (Supplementary Table 10 and Supplementary Note; Methods), indicating the presence of genetic signals not captured by the EDFs (Supplementary Table 10). Furthermore, SPINCs and PLENCs GWAS have higher power (measured by expected chi-square statistics) compared to PCA GWAS[35,36] (Supplementary Tables 11 and 12; Methods).

GWAS on five SPINCs detected 575 independent genome-wide significant (GWS) loci ($r^2 \leq 0.1$ and $P \leq 5 \times 10^{-8}$) after merging hits within 250 kb together (Table 1; Methods). Most GWS loci from SPINCs and EDFs + RSPINCs recover previously known loci[19] (89% for SPINCs and 90% for EDFs + RSPINCs). SPINCs discovered more previously unknown GWS loci (65 of 575, 11%) than EDFs or PCA (Table 1 and Supplementary Note). We observed similarly superior (R)SPINCs performance when compared to a baseline model of nonlinear cubic spline coefficients instead of linear PCs (Supplementary Table 13) and when excluding UKB samples from ref. 19 (Supplementary Table 14). Functional enrichment analysis with GARFIELD[37] shows that these loci are enriched for lung tissue DNase I hypersensitive sites (Supplementary Figs. 20–26 and Supplementary Note) and the EDFs + RSPINCs loci show stronger ontology term enrichments than EDFs loci alone (Extended Data Fig. 8) using GREAT[38]. We performed multiple analyses to ensure that these previously unknown loci were not detected by EDFs or previous work (Supplementary Note and Supplementary Tables 15 and 16).

GWAS on five PLENCs detected 90 independent GWS loci (Table 1; Methods). We compared our PLENCs GWS loci to all cardiovascular function-related loci from the GWAS Catalog[39] (Methods; 520 known independent loci) and GWAS on PPG EDFs. Of the 90 GWS PLENCs loci, 50 (56%) were not previously known (Table 1 and Supplementary Table 17). Functional enrichment analysis showed that PLENCs GWS loci are enriched for fetal heart, heart and blood vessel tissue DNase I hypersensitive sites (Supplementary Figs. 27–33 and Supplementary Note).

## (R)SPINCs improve asthma and COPD PRS over EDFs in UKB

We computed PRSs using BOLT-LMM[35,36] effect sizes for five SPINC and two RSPINC coordinates, in addition to five spirogram EDFs. We treated these sets of PRSs as intermediate genetic scores for lung function. Given a specific trait, a set of such intermediate PRSs and a (small) set of individuals for whom the trait status is available, one can combine the intermediate PRSs into a single trait-specific PRS via a weighted linear sum of the intermediate PRSs. We created disease-specific PRSs for asthma and COPD from the following three sets of intermediate PRSs: (1) five EDFs, (2) five SPINCs and (3) five EDFs plus two RSPINCs. We learned the disease-specific PRS weights within the modeling set ($n = 324{,}958$) of European GIA individuals in UKB using medical-record-based asthma and COPD statuses. To evaluate the performance of each disease-specific PRS, we computed the accuracy of the PRS in a completely separate set of individuals from the European GIA ($n = 110{,}722$) not previously used for model training or GWAS.

We observed that the SPINC asthma PRS stratifies the risk groups more effectively than the EDF PRS on both ends of the risk spectrum (Fig. 4 and Supplementary Table 18). In addition, we observed statistically significant improvements in area under the receiver operating characteristic curve (AUC-ROC), area under the precision-recall curve (AUC-PR) and Pearson correlation using the SPINC PRS (Supplementary Table 18). We observed the same trend for COPD

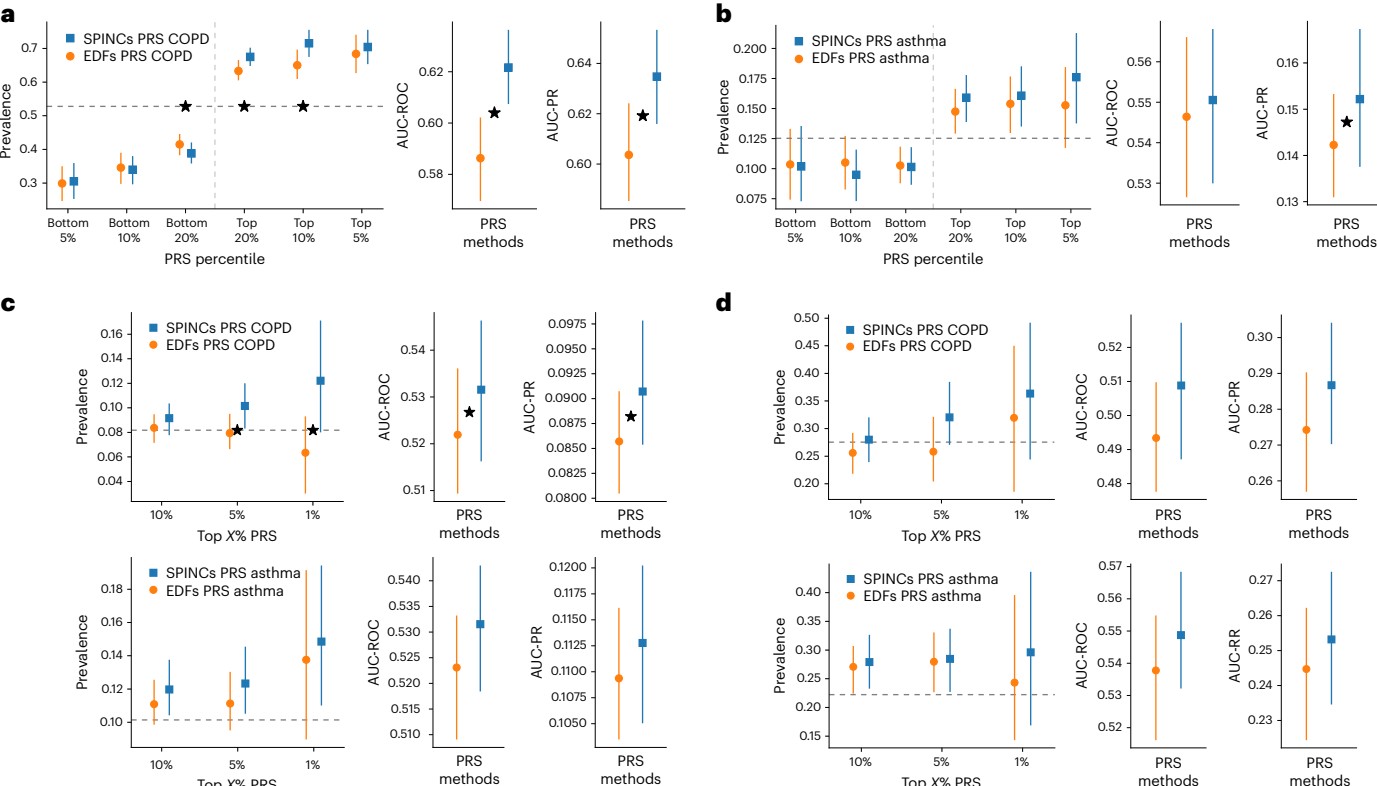

**Fig. 5 | SPINC PRS transferred to multiple independent datasets.** SPINC PRSs (blue) for COPD and asthma generated on UKB are transferred to the following four independent datasets and evaluated against EDF PRSs (orange): COPDGene, eMERGE III, EPIC-Norfolk and Indiana Biobank. **a**, PRS evaluation in COPDGene dataset on COPD. **b**, PRS evaluation in eMERGE III dataset on asthma. **c**, PRS evaluation in EPIC-Norfolk study on COPD and asthma.

**d**, PRS evaluation in Indiana Biobank on COPD and asthma. In all figures, solid vertical intervals represent 95% CIs generated by bootstrapping (300 repetitions), and the center points are the bootstrapping means. The horizontal dashed lines show the total prevalence in the evaluation set. Star symbols indicate a statistically significant difference between the two methods using paired bootstrapping (300 repetitions) with 95% confidence (that is, two-sided $P < 0.05$).

(Fig. 4 and Supplementary Table 19). Furthermore, we observed that the EDF + RSPINC PRS significantly outperforms the EDF PRS on almost all metrics for both asthma and COPD (Fig. 4 and Supplementary Tables 18 and 19). We observed that the SPINC COPD PRS outperforms the $FEV_1$/FVC PRS (Supplementary Table 19) for predicting medical-record-based COPD, despite $FEV_1$/FVC having been shown to be one of the best phenotypes for generating a COPD PRS, even outperforming a PRS created from a GWAS of COPD directly[6]. Finally, we observed that for both diseases, the SPINC and EDF + RSPINC PRSs outperform the PRS generated by baseline methods such as PCA (Supplementary Tables 18 and 19). These results provide further evidence that SPINCs capture more genetic determinants of lung function related to asthma and COPD than the same number of EDFs, and RSPINCs capture additional genetic factors not captured by the EDFs.

We then explored whether disease-specific weights could be learned from a subset of the training data. For both asthma and COPD, the (R)SPINC-based PRS fit with as few as 100 disease cases performed indistinguishably from those trained on the full training data (Fig. 1 (step 3) and Extended Data Fig. 9). Finally, we evaluated PRS generated by GWAS with a cohort-level phenotype adjustment using inverse-normal transformation[40]. While we observed fewer significant differences in this case, SPINCs and EDFs + RSPINCs maintained statistically significant improvement for asthma (Supplementary Fig. 34).

### (R)SPINC PRS transferred to multiple datasets and ancestry

To test the generalizability of our (R)SPINC PRSs to individuals outside the UKB and those of non-European GIA, we transferred our asthma and COPD PRSs to the Genetic Epidemiology of COPD (COPDGene)[41], eMERGE III (dbGaP accession phs001584.v2.p2), European Prospective

Investigation into Cancer in Norfolk (EPIC-Norfolk)[42] and Indiana Biobank datasets[43] (Supplementary Table 20).

For COPDGene, we observed that the SPINC PRS outperforms the EDF PRS on all four evaluation metrics for COPD. In the 'non-Hispanic white' subset ($n = 6,576$), differences in all four metrics were statistically significant (Fig. 5a and Supplementary Table 21). In the 'African American' subset ($n = 3,140$), differences were statistically significant for AUC-ROC and Pearson correlation (Supplementary Table 21). The EDF + RSPINC PRS significantly outperformed the EDF PRS in AUC-ROC and Pearson correlation in 'non-Hispanic white', but did not in the 'African American' subset (Supplementary Table 21 and Supplementary Note).

We also transferred the UKB PRSs to eMERGE III ('white' subset, $n = 8,288$), EPIC-Norfolk (self-reported 'white', $n = 21,010$) and the Indiana Biobank (mostly European GIA, $n = 5,254$; Methods) to evaluate asthma, asthma and COPD and asthma and COPD, respectively. We observed consistent improvement from using SPINC PRSs over EDF PRSs for both COPD and asthma phenotypes for top-percentile prevalences, AUC-ROC and AUC-PR. The improvement was statistically significant for AUC-PR and the top 1% and 5% prevalence in eMERGE III and for AUC-ROC and AUC-PR in EPIC-Norfolk (Fig. 5b–d).

### PLENCs improve hypertension and blood pressure PRS over EDFs

We computed PRSs for the five PLENCs and two RPLENCs plus five PPG EDFs and then used these sets of PRSs as intermediates for constructing cardiovascular function PRSs. We created trait-specific PRSs for HTN and SBP using the REGLE framework (Fig. 1 and Supplementary Table 22). We evaluated HTN and SBP PRSs generated by PLENCs and PPG EDFs in independent datasets (COPDGene, eMERGE III and

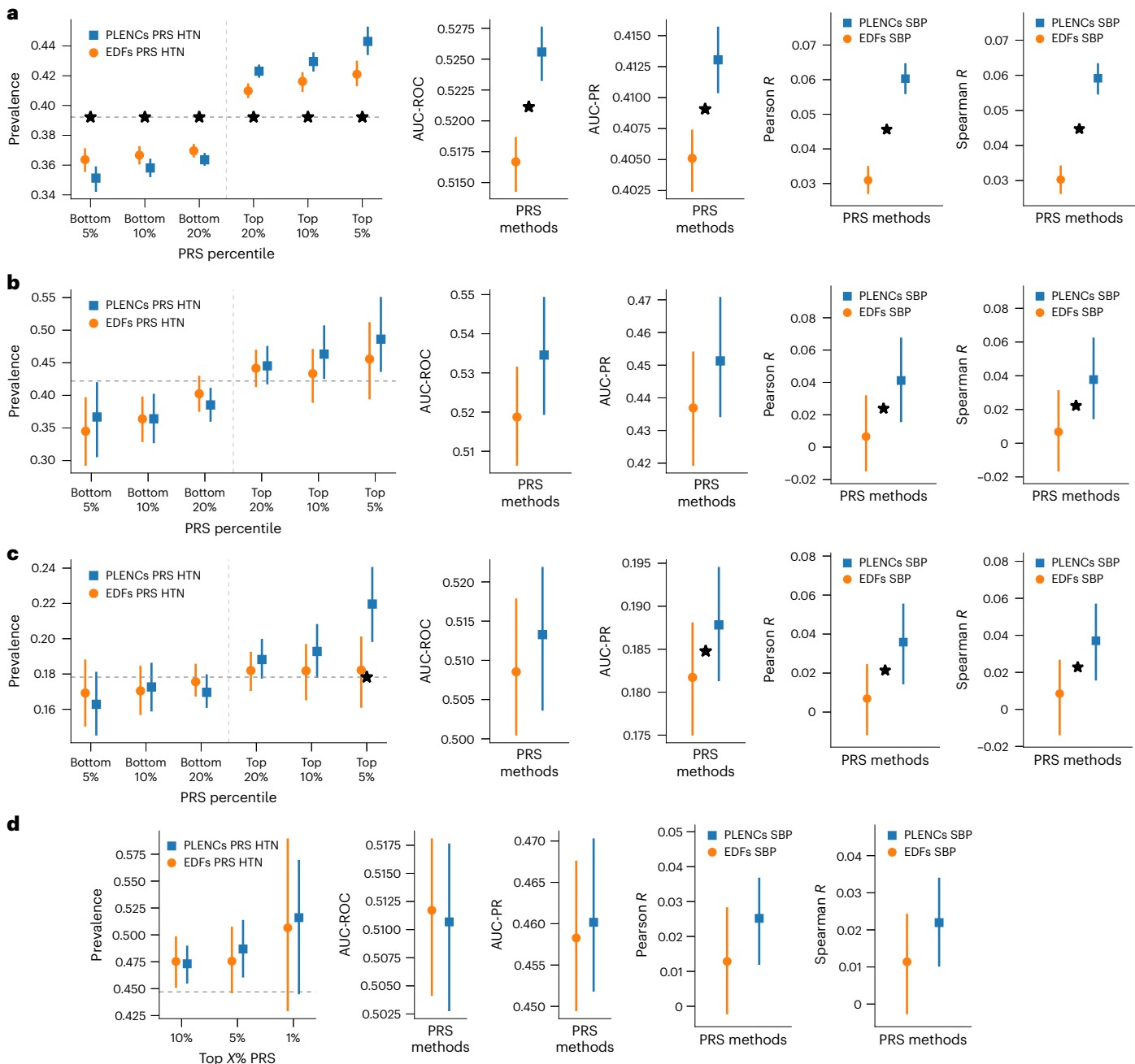

**Fig. 6 | PLENC PRS generated in UKB evaluated in multiple independent datasets.** PLENC PRSs (blue) for HTN and SBP generated on UKB are evaluated in the following four independent datasets against EDF PRSs (orange): UKB, COPDGene, eMERGE III and EPIC-Norfolk. **a**, PRS evaluation in UKB, evaluated in a separate test set not used for GWAS. **b**, PRS evaluation in COPDGene dataset. **c**, PRS evaluation in eMERGE III. **d**, PRS evaluation in EPIC-Norfolk. In all figures, solid vertical intervals represent 95% CIs generated by bootstrapping (300 repetitions), and the center points are the bootstrapping means. The horizontal dashed lines show the total prevalence in the evaluation set. Star symbols indicate a statistically significant difference between the two methods using paired bootstrapping (300 repetitions) with 95% confidence (that is, two-sided $P < 0.05$).

EPIC-Norfolk) in addition to the held-out UKB test set. We did not evaluate cardiovascular PRSs in Indiana Biobank due to the unusually high prevalence of HTN (more than 80%) and blood pressure medication usage by a majority of its population.

We observed a consistent trend of improvement from using PLENC PRSs over EDF PRSs for both HTN and SBP, except for HTN AUC-ROC in EPIC-Norfolk (Fig. 6). Notably, the PLENC PRS for SBP outperformed the EDF PRS for all datasets for both correlation metrics, for example, 2× higher Pearson correlation (6% versus 3%) in the UKB test set (Supplementary Tables 23 and 24), and the differences were statistically significant in three of four datasets.

## High association between REGLE encodings and UKB PRSs

We associated (R)SPINCs and (R)PLENCs with PRSs of 7,145 phenotypes computed by the Pan-UKB consortium (Supplementary Note; Methods). The (R)SPINC PRSs showed a strong correlation with traits previously associated with alterations in lung function, for example, systemic lupus erythematosus[44,45], thyroid dysfunction[46] and gluten-free diet[47] (Supplementary Tables 25–28). (R)PLENCs exhibited significant correlations with different traits, including blood traits, PPG traits, ECG traits, blood pressure and cardiovascular problems (Supplementary Tables 29–32 and Supplementary Note; Methods).

## Discussion

Large biobanks provide unique opportunities to identify the genetic factors underlying complex traits and diseases, but accurate phenotyping[48] remains a core challenge. We proposed a general unsupervised deep learning method, REGLE, to improve genetic discovery for HDCD. We showcased the effectiveness of REGLE for generating encodings of spirograms and PPGs. These are HDCD which, in addition to being routinely measured in clinical settings, can also be measured passively and noninvasively via smartphones. In fact, PPGs are widely collected by popular wearable devices. We demonstrated that the REGLE are both partially interpretable and effective for identifying genetic variants associated with lung and circulatory functions.

Unsupervised learning of HDCD representations for genomic discovery is attractive owing to the difficulty of manually acquiring EDFs at scale. Previous work has explored applying transfer learning[49] and contrastive learning[50] to retinal fundus images, or multimodal autoencoders to cardiac data modalities[51]. A key strength of REGLE is the use of a VAE to generate low-dimensional, nonlinear, disentangled representations. The ability to generate nonlinear representations is desirable for the data applications considered, as the spirogram (Fig. 2) and PPG (Fig. 3) curves seemingly lie close to a low-dimensional manifold and yet are clearly nonlinear. Moreover, VAEs have the following two main advantages over traditional autoencoders: (1) the coordinates of the latent representation are minimally correlated (Extended Data Fig. 5), encouraging them to represent separable biology and increasing power for genetic discovery and PRS (Supplementary Table 33), and (2) the learned representations are stable up to changes in signs or order, which do not affect genetic discovery (Supplementary Note and Extended Data Fig. 3).

To support the principled use of EDFs in modeling, REGLE supports a modification of the VAE in which EDFs are additionally supplied as input to the decoder, implicitly encouraging the encoder to learn features not captured by the EDFs (see Supplementary Note for connection and difference with conditional VAE[52]). Although these models have slightly lower GWAS power and PRS performance compared to nonresidual REGLE (Table 1 and Supplementary Tables 14, 18, 19, 21, 23 and 24), the residual models are intended for capturing variation in HDCD, which is not well-represented by existing EDFs. For example, one of our RSPINCs captures a property of spirometry curves that pulmonologists refer to as 'coving', an indicator of airway obstruction that is not well-represented by the standard EDFs. Moreover, we identified genetic loci associated with this RSPINC (Supplementary Table 16), which may shed light on the mechanisms behind the type of obstruction.

The improved performance of SPINC, EDF + RSPINC, PLENC and EDF + RPLENC PRSs over EDF PRSs provides evidence for the presence of disease-relevant genetic information in HDCD not captured by existing EDFs. Moreover, we developed a label-efficient approach for combining PRSs from GWAS on several learned coordinates. In particular, each coordinate PRS retains its original effect sizes, and a disease-specific PRS is constructed as a learned weighted sum of the handful (that is, five or seven) coordinate PRSs. Because only a minimal number of weights require learning during disease specialization, our premade lung and circulatory system function PRSs can be adapted for risk prediction in new settings with very few disease labels. We hypothesize that unsupervised quantification of other organ systems may be similarly beneficial for improving polygenic prediction across a wealth of diseases. Finally, in cases where labeled data are plentiful, we note that PRS performance can be further improved by jointly estimating disease-specific variant effect sizes across the set of variants associated with our latent coordinates.

There are several limitations to this work. First, we did not directly optimize multiple GWASs for new genomic discovery but used a straightforward (conservative) method to define and merge independent associated loci. A possible extension would be to combine the signals from multiple (R)SPINC and (R)PLENC coordinate GWAS[27]. Second, the VAE objective and, in particular, the reconstruction error are not necessarily optimal for genetic analyses, and explicitly incorporating an objective to maximize the heritability of the learned representation may be a fruitful line of future research[53]. Third, we did not fully optimize model architecture and training strategies specifically for genomic discovery (Supplementary Note). Fourth, we generated individual-level spirogram representations from the first measurement, despite some individuals having up to three acceptable blows. Integrating all acceptable blows from an individual could produce a more comprehensive representation of their lung function[54]. Fifth, REGLE was trained on spirograms and PPG obtained from the UKB only; thus, (R)SPINCs and (R)PLENCs representations may not generalize well to other datasets. One needs additional datasets with the same data modality to investigate the generalizability of the encodings. Finally, model training was performed exclusively on individuals of European GIA. While PRS evaluation was performed on multiple datasets and ancestries, the impact of ancestry-specific model training was not explored.

Despite these limitations, REGLE provides a mechanism for identifying genetic influences on organ function in the absence of labeled data and naturally admits to incorporating expert features into the model. It also provides a method to create disease/trait-specific PRS with very few labels (that is, in the order of hundreds). As biobanks with rich imaging, activity monitoring, medical records and paired genetic data continue to grow, we anticipate that this or similar methods will be increasingly used to further elucidate the genetic underpinnings of human traits and diseases.

## Online content

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

[1]Google Research, Cambridge, MA, USA. [2]Google Research, Mountain View, CA, USA. [3]Department of Electrical and Computer Engineering, Northeastern University, Boston, MA, USA. [4]Channing Division of Network Medicine, Brigham and Women's Hospital, Boston, MA, USA. [5]NIHR Biomedical Research Centre at Moorfields Eye Hospital and University College London (UCL) Institute of Ophthalmology, London, UK. [6]MRC Epidemiology Unit, University of Cambridge, Cambridge, UK. [7]Department of Medical and Molecular Genetics, Indiana University School of Medicine, Indianapolis, IN, USA. [8]Verily Life Sciences, South San Francisco, CA, USA. [9]Division of Cardiology, Department of Medicine, Indiana University School of Medicine, Indianapolis, IN, USA. [10]Division of Pulmonary and Critical Care Medicine, Brigham and Women's Hospital, Boston, MA, USA. [11]Harvard Medical School, Boston, MA, USA. [12]Present address: Insitro, South San Francisco, CA, USA. [13]These authors jointly supervised this work: Cory Y. McLean, Farhad Hormozdiari. ✉e-mail: tedyun@google.com; cym@google.com; fhormoz@google.com

## Methods

All relevant ethical guidelines have been followed for this research, and any necessary institutional review board (IRB) and/or ethics committee approvals have been obtained. Advarra IRB (Columbia, MD) waived ethical approval for this work involving de-identified medical imagery and metadata under 45 Code of Federal Regulations 46. Work related to genomics data was additionally reviewed by the respective data sources—UKB, COPDGene, eMERGE III, EPIC-Norfolk and Indiana Biobank. This research has been conducted using the UKB resource under application 65275.

### UKB data preparation for spirograms

Spirograms from UKB were sourced from the data field 3066, which contains the volume in milliliters of exhalation at 10-ms intervals (volume–time curve), and were preprocessed closely following the procedures in ref. 6. To generate flow–time curves, we approximated the first derivative of volume with respect to time by taking a finite difference in the volume–time curves. We normalized the volume–time and flow–time curves to 1,000 time points by either truncating longer curves or by right-padding shorter curves with zero (for flow–time curves) or the final value (for volume–time curves), and removed $FEV_1$, FVC and PEF values in the extreme tail (top or bottom 0.5%) of the observed values and all blows that failed to meet the acceptability provided by UKB data field 3061. We used the first acceptable blow of an individual when there was more than one. In addition, we dropped all flow curves whose values don't fall in (−10, 20), all volume curves whose values are not in (−5, 10) and all flow curves in which the proportion of nonzero values is less than 20%. Finally, we generated flow–volume curves from volume–time and flow–time curves by interpolating 1,000 evenly spaced volume values between 0 and 6.58 l (the maximum observed volume in the dataset).

We then subdivide all European GIA individuals processed this way into an 80% training set and a 20% validation set similar to ref. 6. After additionally removing related individuals, there are 259,692 individuals in the training set and 65,266 individuals in the validation set (Extended Data Fig. 1).

Asthma and COPD statuses were determined by medical records using self-report, International Classification of Diseases (ICD)-9 and ICD-10 codes as defined in ref. 6.

### UKB data preparation for PPGs

PPGs from UKB were sourced from the data field 4205, which contains the arterial stiffness pressure curve. Each waveform is actually a single pulse with 100 points. Then we computed the minimum, maximum, mean and median distribution values of PPG. We keep the PPG when all four statistics fall in 0.1 and 99.9 percentiles of the related statistics values of all PPGs. We then subdivide all European GIA individuals processed this way into an 80% training set and a 20% validation set. After additionally removing related individuals, there are 112,730 individuals in the training set and 28,545 individuals in the validation set (Extended Data Fig. 4).

HTN status was determined by medical records using self-report, ICD-9 codes (401.* and 405.*) and ICD-10 codes (I10 and I15.*). SBP was determined by automated reading, and data field 4080 was used in UKB.

### Convolutional VAE model architecture and training

To generate SPINCs, we encode the flow–time and volume–time curves. In our VAE, we use one-dimensional (1D) convolutional layers to use the temporal context of this time series, encoding the two curves in two channels. In the encoder, we first apply three 1D convolutional layers, each followed by max pooling. We use three fully connected layers to generate the mean and variance of the bottleneck layer. We use five latent dimensions, identical to the number of EDFs, and each latent coordinate is sampled from the Gaussian distribution with the learned means and variances. The decoder architecture is a mirror image of the encoder. We start with three fully connected layers followed by transpose convolution layers, each prepended by an upsampling layer (see Extended Data Fig. 10 and SPINCs model architecture in Supplementary Note for full details).

For RSPINCs, we encode the flow–volume curve alone, and we apply the same sequences of convolutional and fully connected layers as we did for SPINCs, while using only two latent dimensions in this case. We chose to use two latent dimensions for the encoder based on REGLE's strong reconstruction performance (Extended Data Fig. 2) while maintaining a comparable number of total latent dimensions to SPINCs. Notably, we use a modified VAE architecture to concatenate the five EDFs directly to the sampled output of the bottleneck layer (the layer right before the decoder) to learn only the residual signals not represented by the EDFs (Fig. 2a). As a result, the encoder output dimension is 2, while the decoder input has dimension 5 + 2 = 7 (see Extended Data Fig. 10 and RSPINCs model architecture in Supplementary Note for full details).

For PLENCs, we encode the PPG curves. In our VAE, we use 1D convolutional layers to use the temporal context of this time series. In the encoder, similar to SPINCs, we first apply three 1D convolutional layers, each followed by max pooling, and use three fully connected layers to generate the mean and variance of the bottleneck layer. We use five latent dimensions, identical to the number of EDFs, and each latent coordinate is sampled from the Gaussian distribution with the learned means and variances. The decoder architecture is a mirror image of the encoder. We start with three fully connected layers followed by transpose convolution layers, each prepended by an upsampling layer. RPLENCs are generated similarly to RSPINCs where we inject five EDFs directly into the sampled output of the bottleneck layer (see Extended Data Fig. 10 and PLENCs model architecture and RPLENCs model architecture in Supplementary Note for full details).

All models are trained using the standard VAE loss function consisting of the reconstruction loss and the (rescaled) Kullback–Leibler (KL) divergence loss. For RSPINCs, the KL divergence loss is only applied to the learned encodings, not to the injected EDFs. For optimization, the Adam optimizer[55] is used with varying learning rates and batch sizes. No learning rate scheduler was used. After training for 100 epochs, the final learning rate and batch size values (hyperparameters) for (R)SPINCs and PLENCs were chosen to minimize the VAE loss in the validation set (Supplementary Note and Supplementary Table 1).

After training SPINCs, RSPINCs and PLENCs models, we use the encoders of the trained models to generate the encodings for each individual, using the mean value of the learned Gaussian distribution of the encodings. It is worth mentioning that the learned variance for VAE is not used.

All models were implemented in TensorFlow V2 (ref. 56).

### Principal components (PCs) and cubic spline coefficients

As baseline methods for dimensionality reduction, we performed PCA and cubic spline fitting on spirograms. For PCA we concatenated volume–time and flow–time curves and used both as inputs, while for cubic spline fitting, we used only volume–time curves as cubic splines perform better for 'smoother' curves. To match the number of EDFs and the dimension of SPINCs, we generated five PCs and five cubic spline coefficients. We used one knot and cubic curves for spline fitting to generate exactly five coefficients, where the knot position was chosen at the 20% position to better capture the complexity at the beginning of the volume–time curves. We used scikit-learn (v1.0.2) for PCA and SciPy (v1.9.3) for spline fitting.

### Phenotypic correlation analysis

To residualize EDFs and/or covariates from (R)SPINCs and PLENCs, we used ordinary least squares linear regression. To compute the correlation of the EDFs-and-covariates-residualized (R)SPINCs and PLENCs with the tabular fields in UKB, we first preprocessed the tabular fields

to remove special codes, normalize, impute and aggregate the values and then finally transformed the categorical fields into one-hot encodings. For each correlation analysis between a feature and one of the (R)SPINCs and PLENCs, we computed the Pearson correlation and the P value with a two-sided alternative hypothesis.

## Survival analysis

We performed an analysis of overall survival for European GIA individuals in the spirometry ($n = 65,266$) and PPG ($n = 28,545$) validation sets using the time from first assessment (field 53) to death from any cause (field 40000). Participants who were not known to have died were right-censored at the date of UKB data ingestion (18 December 2020). We quantified the association between overall survival and each SPINC, RSPINC, PLENC, RPLENC and EDF per s.d. using the hazard ratio, which was estimated from a Cox proportional hazards model adjusting for age and sex. The proportional hazards assumption, with respect to each SPINC, RSPINC, PLENC, RPLENC and EDF, was assessed using the Schoenfeld residual test. After stratifying patients into quartiles using each SPINC, RSPINC, PLENC, RPLENC or EDF coordinate, the overall survival curves were constructed using the standard Kaplan–Meier estimator with bootstrapped 95% CIs.

## GWAS and PRSs

GWAS on all spirograms EDFs, SPINCs and RSPINCs were performed using BOLT-LMM (v2.3.6)[35,36], adjusting for age, sex, age$^2$, age × sex, height, height$^2$, body mass index, smoking status, the number of packs of cigarettes smoked per year, the type of genotyping array and the top 15 genetic PCs as covariates. GWAS on all PPG EDFs, PLENCs and RPLENCs were performed using BOLT-LMM[35,36], adjusting for the same covariates as SPINCs while excluding smoking status and the number of packs of cigarettes smoked per year.

All GWAS were restricted to European GIA individuals to minimize confounding. For QC we kept variants with minor allele frequency ≥0.001, imputation INFO score ≥0.8, missing call fraction ≤0.05 and Hardy–Weinberg equilibrium P value ≥ $10^{-10}$, among all genotyped and imputed variants provided by UKB. After GWAS, we performed Stratified Linkage Disequilibrium Score Regression[57] to estimate SNP heritability and detect potential confounding. GWS 'hits' were defined as the most significant variants with $P \leq 5 \times 10^{-8}$ and independent at $r^2 < 0.1$ using the PLINK `--clump` command. A reference panel for linkage disequilibrium (LD) calculation contained 10,000 unrelated European GIA samples from the UKB. Significant 'loci' were created based on the span of reference panel SNPs in LD ($r^2 \geq 0.1$) with the hits. Loci separated by fewer than 250 kb were subsequently merged.

While performing GWAS, PRSs for all traits were computed using the `--predBetasFile` option of BOLT-LMM. While GWAS was performed on individuals with valid spirometry measurements, we evaluated the performance of the PRS in a separate set of individuals not used for GWAS. More specifically, we use the following model to predict the $i$th individual phenotype. To estimate the PRS weight of the i-th individual for the $k$th latent embedding ($s_{ik}$), we use the following model:

$$s_{ik} = \sum_{j=1}^{M} g_{ij} \hat{\beta}_{jk},$$

where $g_{ij}$ is the $i$th individual genotype at the $j$th variant and $M$ is the total number of variants or SNPs. $\hat{\beta}_{jk}$ is the effect size estimated by BOLT-LMM for $j$th variant and $k$th latent dimension. Next, we estimate the $i$th individual phenotype of interest as follows:

$$y_i = \sum_{j=1}^{T} s_{ik} w_k,$$

where $w_k$ is the linear effect size estimated via an in-house linear model. In all cases that we have five latent embeddings, we set $T$ to 5.

To determine the known lung function loci from previous literature, we extracted all significant loci from ref. [19] by downloading the full GWAS summary statistics and merging hits using the exact same criteria and P value threshold described above, and searched for lung function-related keywords in the NHGRI-EBI GWAS Catalog (`v1.0.2-associations_e106_r2022-07-09`)[39]. We used the following keywords (case insensitive) for the catalog search: 'asthma', 'chronic obstructive pulmonary disease', 'copd', 'expiratory flow', 'fev1', 'forced expiratory', 'forced vital capacity' and 'lung function'. To determine the known cardiovascular function loci from previous literature, we used the following keywords (case insensitive) for the NHGRI-EBI GWAS Catalog search: 'arrhythmia', 'afib', 'atrial fibrillation', 'coronary artery disease', 'stroke', 'heart attack', 'myocardial infarction', 'heart failure', 'mace' and 'rheumatic heart disease'.

## Statistical power via expected chi-square statistics

We used expected chi-square statistics ($E(\chi^2)$) for all variants or known GWAS Catalog variants related to lung or cardiovascular traits as a measure of statistical power[35,36]. We computed the chi-square statistics for a given variant for a set of phenotypes with extremely low correlation (for example, methods such as PCA and REGLE) by summing the $\chi^2$ for all phenotypes while incorporating the degrees of freedom equal to the number of phenotypes (for example, degrees of freedom of five for SPINCs and PLENCs, five for PCA with five PCs and four when we have used four PCs). Then, we computed the expected chi-square statistics ($E(\chi^2)$) for all or a subset of variants (for example, variants associated with lung function in the GWAS Catalog).

## Respiratory diseases and cardiovascular traits on multiple datasets

**COPDGene dataset.** COPDGene is a study of 10,300 current and former smokers with and without COPD, self-reported non-Hispanic white and African Americans, without known lung diseases other than COPD and asthma (dbGaP accession: phs000179.v6.p2). Additional study details, the study protocol and details of genotyping have been previously published[41,58], and additionally detailed at copdgene.org. We used the provided variant calls in VCF files and imputed the variants to the Haplotype Reference Consortium (HRC) reference panel using Michigan Imputation Server[59], resulting in 39,127,678 total variants. COPD cases were determined using the Global Initiative for Chronic Obstructive Lung Disease (GOLD) criteria, where GOLD stage 2 or higher was considered as cases. Among 6,576 non-Hispanic white individuals, we had access to 1,131 (17%) asthma cases and 2,781 (42%) COPD cases, and the rest of the individuals were used as controls. Meanwhile, among 3,140 African American individuals, 760 (24%) were asthma cases and 802 (26%) were COPD cases. We used the blood pressure measurements and the 'high blood pressure' variable included in the dataset to define SBP and HTN traits.

**EPIC-Norfolk dataset.** The EPIC-Norfolk is a general population-based cohort study of men and women aged 40–79 years living in Norfolk, UK and recruited from general practices between 1993 and 1997. EPIC-Norfolk cohort participants were linked annually to nationally held hospital records and death certificates from 1999 to 2019 using UK National Health Service numbers. COPD was defined as any hospital admission or cause of death coded 490–492, 494–496 (ICD-9) or J40–J44, J47 (ICD-10). Asthma was similarly defined using codes 493 (ICD-9) or J45, J46 (ICD-10). HTN was defined using hospital records and death certificates for ICD codes 401.*, 405.* (ICD-9) and I10, I15.* (ICD-10). The SBP was determined using the continuous SBP from the EPIC-Norfolk health examination at baseline, which is the time point with the highest number of individuals. In a small set of participants who do not have a baseline blood pressure measurement, we used blood pressure measured at the earliest subsequent health examination. Blood pressure was measured at two time points during the examination, with the mean used for analysis.

**eMERGE III dataset.** We use the following five consent groups that do not require IRB approval: General Research Use (GRU), Health/Medical/Biomedical-Genetic Studies (HBM-GSO), Health/Medical/Biomedical (HMB), Health/Medical/Biomedical (MDS) HMB-MDS and Health/Medical/Biomedical (PUB, GSO) (HMB-PUB-GSO; dbGaP accession: phs001584.v2.p2). We have access to 1,038 asthma cases and 7,250 controls for European GIA, while in the case of African GIA, we have access to 649 asthma cases and 1,367 controls. We used the 39,131,578 variants that are imputed to the HRC reference provided by dbGaP[60]. Asthma and SBP traits were defined using the corresponding variables in the dataset. HTN was defined using two variables, 'CASE_CONTROL_CKD_T2D_HTN' and 'CASE_CONTROL_CKD_T2D', where the individuals in the former group but not in the latter group were defined as the HTN cases. Note that this limited our analysis to hypertensive individuals without chronic kidney disease or type 2 diabetes.

**Indiana Biobank dataset.** The Indiana Biobank is a state-wide collaboration that provides centralized processing and storage of specimens that are linked to participants' electronic medical information via the Regenstrief Institute at Indiana University. COPD was diagnosed by using ICD-9 (491, 492 and 496) and ICD-10 (J41, J42, J43 and J44). Asthma was diagnosed by using ICD-9 (493) and ICD-10 (J45 and J46). Cases were defined as having at least one in-patient diagnosis or two out-patient diagnoses. Those participants who did not have any diagnoses were defined as controls. Thus, we have 1,445 COPD cases and 3,808 controls, while we have 1,171 asthma cases and 4,083 controls. Among 5,253 individuals for COPD evaluation, 3,797 were of European GIA, 1,371 were of African GIA and 85 were of Hispanic ancestry. Among 5,254 individuals for asthma evaluation, 3,805 were of European GIA, 1,362 were of African GIA and 87 were of Hispanic ancestry. Indiana Biobank samples used in this study were genotyped using the Illumina Infinium Global Screening Array by Regeneron. SNPs with missing rate >5%, minor allele frequency ≤1% and Hardy–Weinberg equilibrium $P$ value $<1 \times 10^{-10}$ among cases and $<1 \times 10^{-6}$ in controls were excluded as reported previously[43]. Genotyping data were imputed to 1000 Genomes using the Michigan Imputation Server[59]. Imputed variants with $r^2 < 0.30$ and minor allele frequency $< 1\%$ were excluded. PLINK[61,62] was used to calculate PRS using imputation dosages.

### Functional significance of discovered loci
We ran GREAT (v4.0.4)[38] on the human GRCh37 assembly to perform functional enrichment analysis of SPINCs, RSPINCs, PLENCs, RPLENCs and EDF loci. We used the default 'basal + extension' region–gene association rule with 5 kb upstream, 1 kb downstream, 1,000 kb extension and curated regulatory domains included. Furthermore, we ran GARFIELD (v2)[37] with default parameters to perform tissue-specific analysis where we used 424 DNase I hypersensitive site hotspot annotations provided by the GARFIELD authors[37].

### Genetic phenome-wide association study
To compute PheWAS, we downloaded GWAS summary statistics for 7,221 phenotypes from the Pan-UKB consortium (20200615 release; https://pan.ukbb.broadinstitute.org). After restricting to phenotypes that contained European GIA statistics and did not persistently fail in LD clumping, we were left with 7,145 pruning + thresholding (P + T) PRSs generated by PLINK (https://www.cog-genomics.org/plink1.9) using the `--clump` command with an index variant significance threshold of $5 \times 10^{-8}$ and LD threshold of 0.1, with LD computed from a random subset of 10,000 European GIA individuals in UKB.

SPINCs, RSPINCs and PLENCs P + T PRSs were computed analogously to the Pan-UKB PRSs. We computed the Pearson correlations between the PRSs derived from latent dimensions and the PRSs derived from Pan-UKB phenotypes and the $P$ values with a two-sided alternative hypothesis.

### Statistics and reproducibility
All codes necessary to reproduce the results in this work are available on our GitHub repository. No statistical method was used to predetermine the sample size. The experiments were not randomized. The investigators were not blinded to allocation during experiments and outcome assessment. We removed samples with no valid blows from our spirogram analyses. To QC the blows, we drop any blow if one of $FEV_1$, FVC and PEF values is in the extreme tail (top or bottom 0.5%). We dropped all flow curves whose values don't fall in (−10, 20), all volume curves whose values are not in (−5, 10) and all flow curves in which the proportion of nonzero values is less than 20%, assuming these blows are likely noisy. We also removed blows that failed the acceptability (valid) provided by UKB. We treated a blow as valid if the value for the UKB field 3061 is 0 or 32. For PPG analysis, we removed outliers defined by any of the four statistics (for example, minimum, maximum, mean and median) falling in the extreme tails of the distribution (top or bottom 0.1%).

### Reporting summary
Further information on research design is available in the Nature Portfolio Reporting Summary linked to this article.

### Data availability
(R)SPINCs and (R)PLENCs values of UKB individuals will be returned to UKB and will be made available by UKB. GWAS summary statistics for SPINCs, RSPINCs, PLENCs, RPLENCs and EDFs are freely available on Google Cloud Storage at https://console.cloud.google.com/storage/browser/brain-genomics-public/research/regle and were also submitted to the GWAS Catalog for wider availability (under GCP000877; 24 consecutive study accessions from GCST90399850 to GCST90399873). All variant weights for PRSs used in this paper, including the intermediate EDFs and encodings PRSs and the disease-specific PRSs for asthma, COPD, HTN and SBP, are also available in the same Google Cloud Storage location under the `prs_model` directory.

### Code availability
Open-source code and trained model weights are available at https://github.com/Google-Health/genomics-research under the `regle` directory[63].

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

## Acknowledgements

We thank all participants, dataset creators and maintainers of UKB, COPDGene, eMERGE III, EPIC-Norfolk and Indiana Biobank. Full acknowledgement for each dataset can be found in Supplementary Note—Dataset acknowledgment. We also thank N. Furlotte for helpful discussions. D.H. is supported by National Institutes of Health (NIH) 2T32HL007427-41 from the National Heart, Lung, and Blood Institute. T.-H.S.-A. and D.L. are supported by the Indiana Clinical and Translational Sciences Institute funded Indiana Biobank, in part by UL1TR002529 from NIH, National Center for Advancing Translational Sciences, Clinical and Translational Sciences Award. T.-H.S.-A. and D.L. did not receive any specific funding for this work. A.P.K. is supported by a UK Research and Innovation Future Leaders Fellowship, an Alcon Research Institute Young Investigator Award and a Lister Institute for Preventive Medicine Award. R.L. and A.P.K. were supported by the National Institute for Health and Care Research Biomedical Research Centre at Moorfields Eye Hospital and the University College London Institute of Ophthalmology. The EPIC-Norfolk study has received funding from the Medical Research Council (MR/N003284/1 and MC-UU-12015/1) and Cancer Research UK (C864/A14136). The genetics work in the EPIC-Norfolk study was funded by the Medical Research Council (MC-PC-13048). We are grateful to all the participants who have been part of the project and to the many members of the study teams at the University of Cambridge who have enabled this research. B.D.H. was supported by R01HL162813, R01HL155749, R01HL160008, U01HL089856 and a research grant from the Alpha-1 Foundation. M.H.C. was supported by R01HL162813, R01HL153248, R01HL149861 and R01HL147148. The content is solely the responsibility of the authors and does not necessarily represent the official views of the NIH. This study was funded by Google. The funders had no role in study design, data collection and analysis, decision to publish, or preparation of the paper.

## Author contributions

F.H. conceived the study. T.Y. and F.H. designed the study in discussion with B.B. and C.Y.M. T.Y., J.C., B.B., Z.R.M., D.H., R.L., D.L., J.B., T.-H.S.-A., Y.Z., A.P.K., C.Y.M. and F.H. performed experiments and analyzed results. B.D.H. and M.H.C. analyzed results and provided clinical expertise. H.Y. and A.C. performed project administration. T.Y., M.H.C., C.Y.M. and F.H. wrote the paper with contributions from all authors. All authors edited, reviewed, and approved the final version of the paper.

## Competing interests

T.Y., J.C., B.B., Z.R.M., H.Y., J.B., Y.Z., A.W.C., C.Y.M. and F.H. are current or former employees of Google or Verily and own Alphabet stocks. A.P.K. has acted as a paid consultant or lecturer to AbbVie, Aerie, Allergan, Google Health, Heidelberg Engineering, Novartis, Reichert, Santen, Thea and Topcon. B.D.H. received grant support from Bayer and has previously received an honorarium from AstraZeneca for an educational lecture. The remaining authors declare no competing interests.

## Additional information

**Extended data** is available for this paper at https://doi.org/10.1038/s41588-024-01831-6.

**Correspondence and requests for materials** should be addressed to Taedong Yun, Cory Y. McLean or Farhad Hormozdiari.

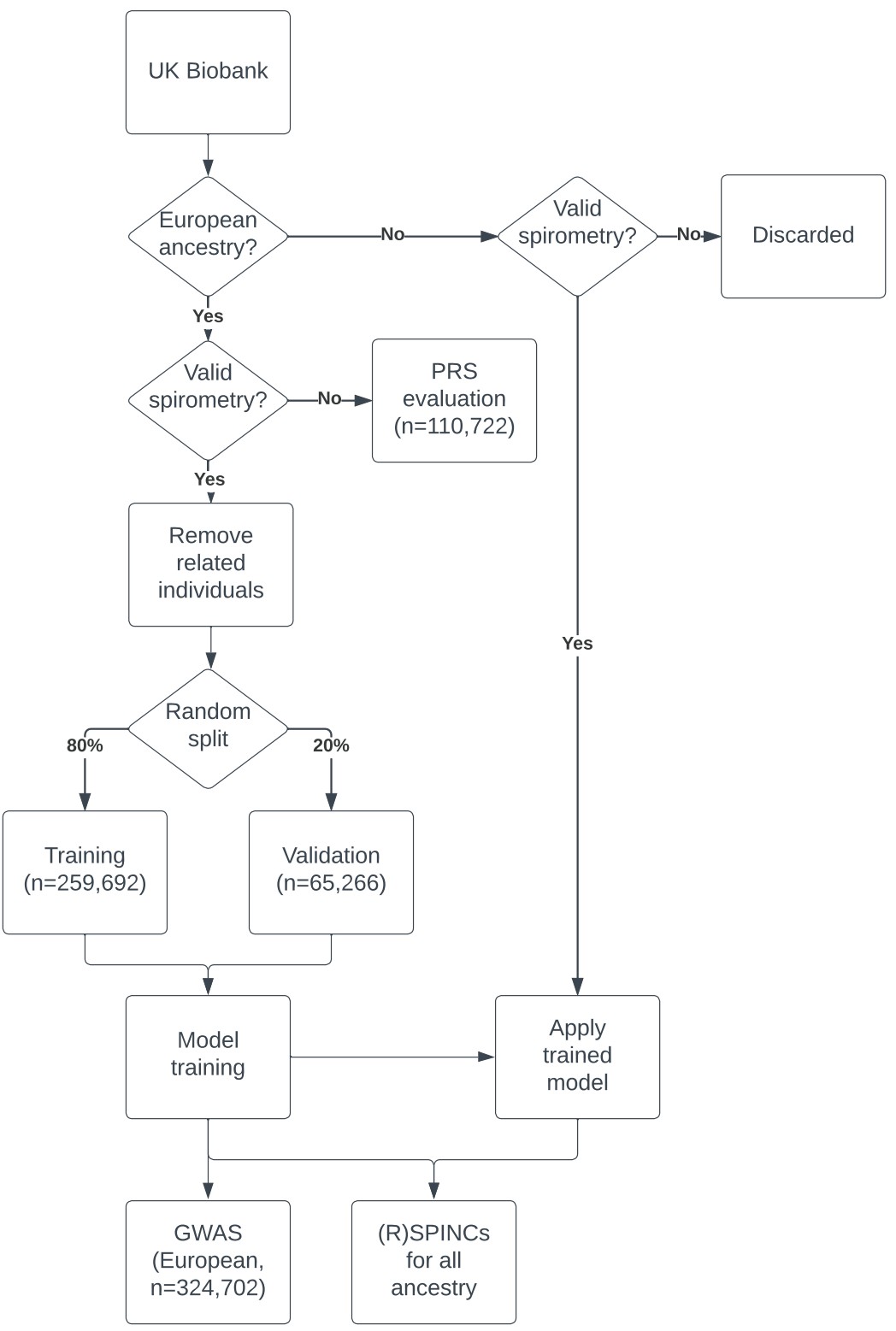

**Extended Data Fig. 1 | An overview of UK Biobank spirograms used in this study.** Our initial dataset consists of all European GIA (genetically inferred ancestry) in UK Biobank (n = 435,766). We considered all individuals with valid spirograms as modeling datasets (n = 325,027), and individuals with invalid spirograms are used as PRS holdout sets. The PRS holdout set is from the European GIA individuals who are not used in the ML modeling and in the GWASs (n = 110,739). We split the ML modeling set into training (80%) and validation (20%) sets. We used all individuals in modeling set for GWAS analysis and generated (R)SPINCs for individuals with valid spirometry in all ancestry.

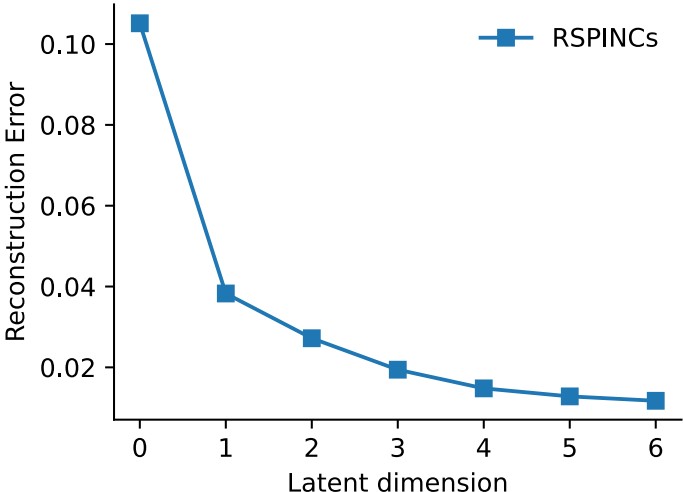

**Extended Data Fig. 2 | Reconstruction error using RSPINCs with varying latent dimensions.** Note that all RSPINC models include spirogram EDFs (dim = 5), so the total number of inputs used for reconstructing curves is 5 plus the latent dimension. The latent dimension of zero in the plot implies bypassing the encoder and the sampling layer of the VAE and solely using EDFs to reconstruct spirograms. For comparison, using the training set average curve for all evaluation set individuals would yield the reconstruction error of 2.057727, 20× higher than any data point in this figure.

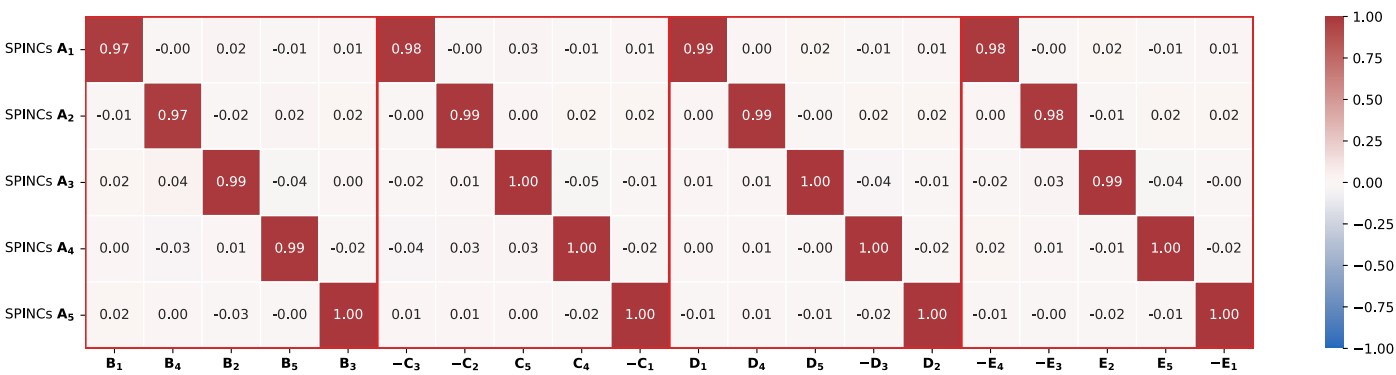

**Extended Data Fig. 3 | SPINCs trained with different random seeds.** Five SPINCs are trained from an identical model using different random seeds to initialize the training: models A, B, C, D and E. The Pearson correlations between the coordinates of model A and the coordinates of models B, C, D and E are displayed as a heatmap. Note the order and signs of the coordinates of models B, C, D and E are permuted and flipped as indicated in their x-axis labels to maximize correlation with coordinates of model.

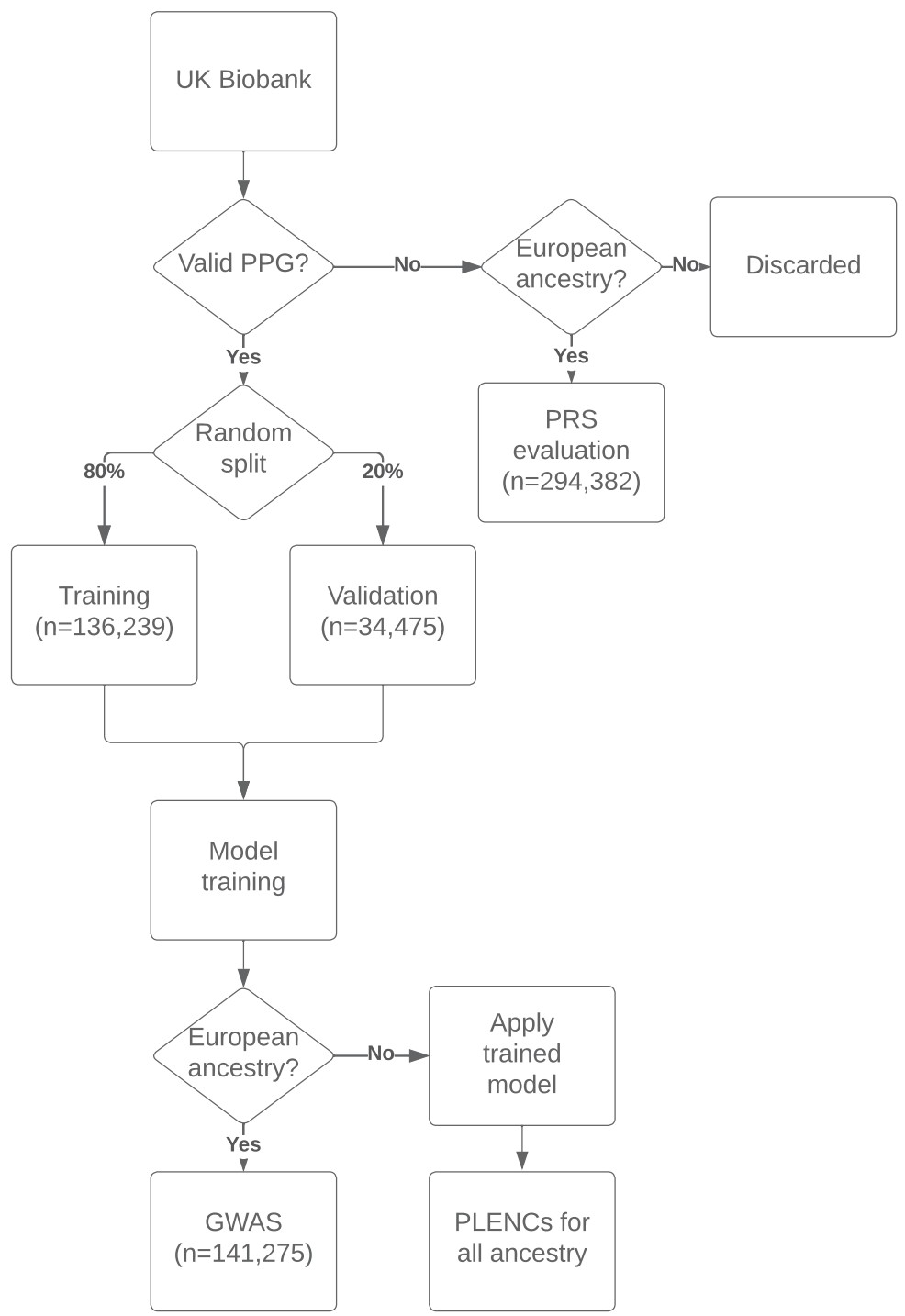

**Extended Data Fig. 4 | An overview of UK Biobank PPGs used in this study.** We considered all individuals with PPGs as modeling dataset (n = 170,714), and individuals with invalid spirograms are used as PRS holdout set. The PRS holdout set is from the European GIA individuals who are not used in the ML modeling. We split the ML modeling set into training (80%) and validation (20%) sets. We used all European GIA individuals in modeling set for GWAS analysis and generated PLENCs for individuals with valid PPGs in all ancestry.

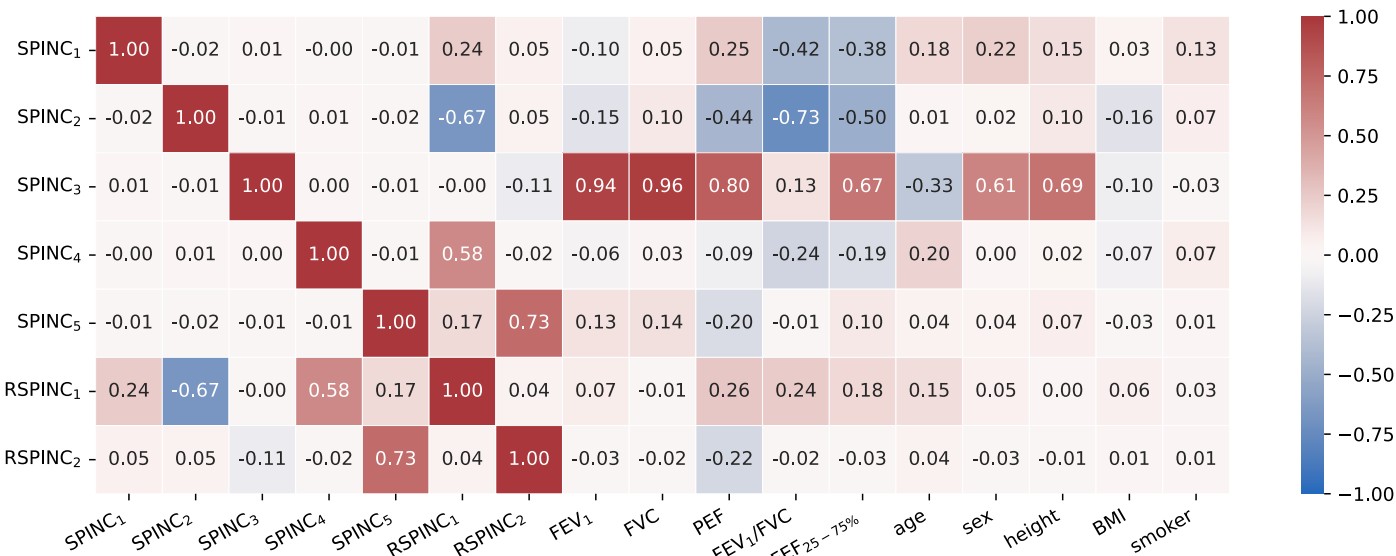

**Extended Data Fig. 5 | Correlation between SPINCs and RSPINCs coordinates and manual metrics and covariates.** Pearson correlation between the coordinates of SPINCs (dim = 5), RSPINCs (dim = 2) and the manual spirometry metrics (for example, $FEV_1$) and other covariates (for example, age and sex).

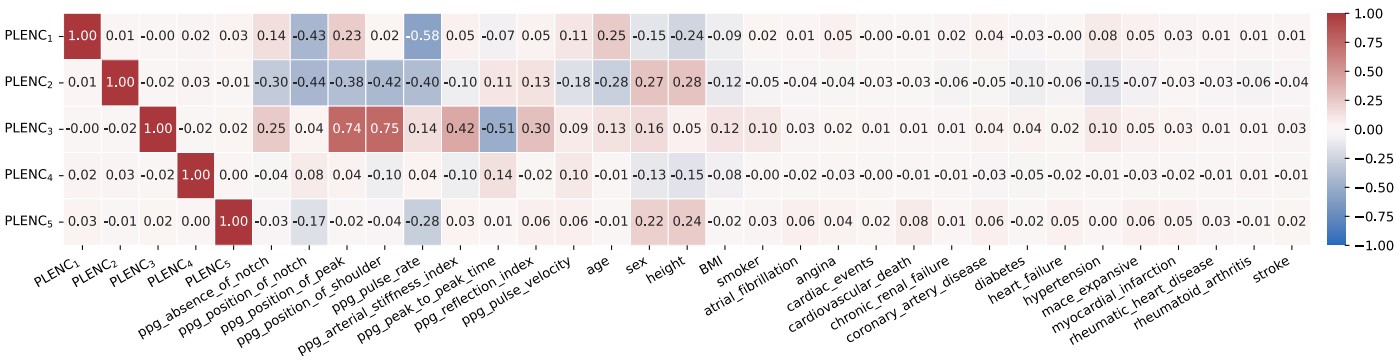

**Extended Data Fig. 6 | Correlation between PLENCs coordinates and manual metrics, covariates and cardiovascular diseases.** Pearson correlation between the coordinates of PLENCs (dim = 5) and the manual PPG metrics (for example, notch position) and other covariates (for example, age and sex).

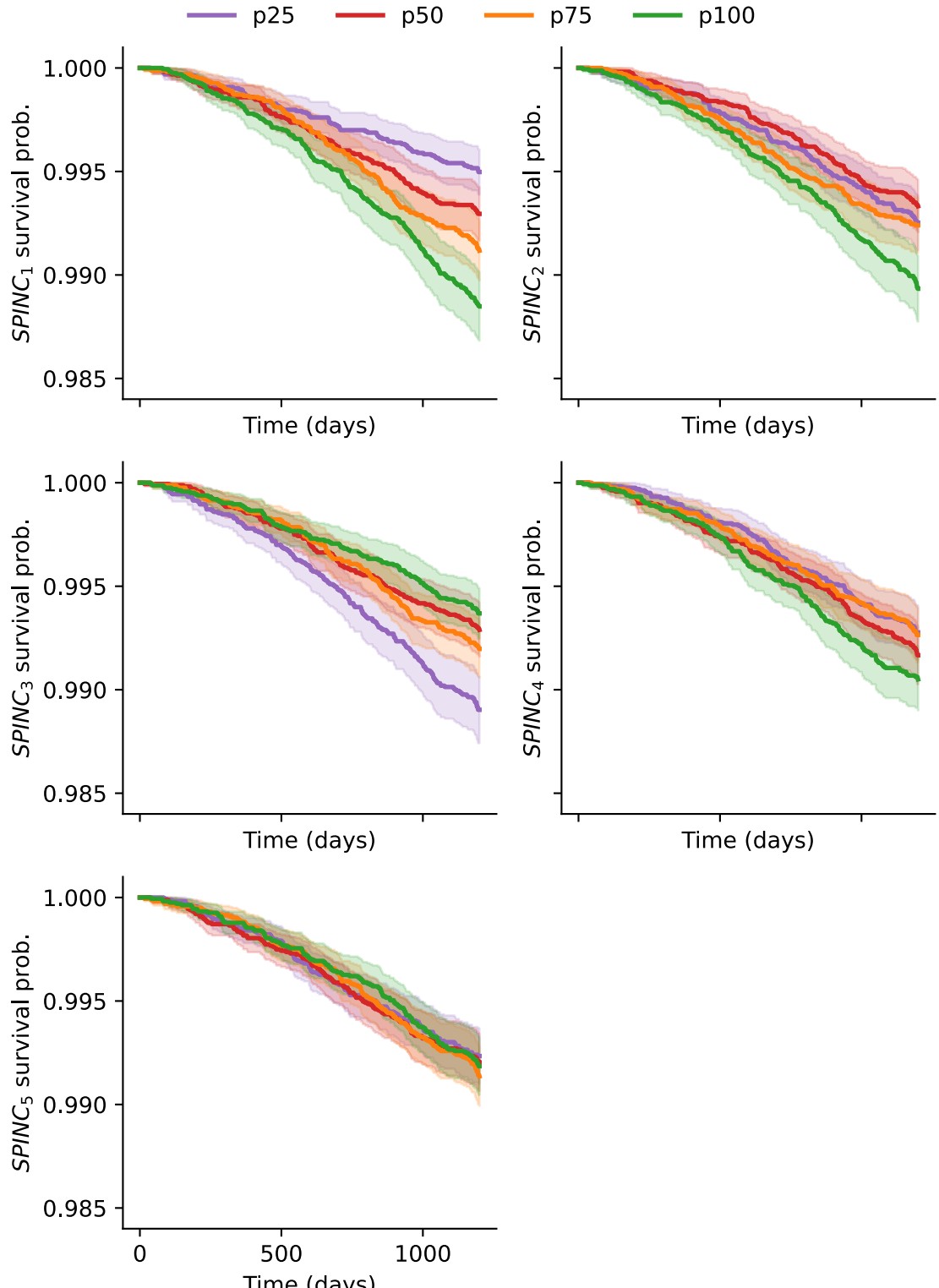

**Extended Data Fig. 7 | SPINCs Kaplan–Meier curves.** Kaplan–Meier curves estimating the overall survival (OS) function for European GIA individuals in the validation dataset (n = 65,266). Individuals were stratified into quartiles using each SPINC coordinate (for example, 'p25' denotes the bottom quartile), and OS curves were constructed using the standard Kaplan–Meier estimator with bootstrapping. The center lines are the means and the error bands are bootstrapped 95% confidence intervals. See Supplementary Table 9 for the corresponding hazard ratios per standard deviation.

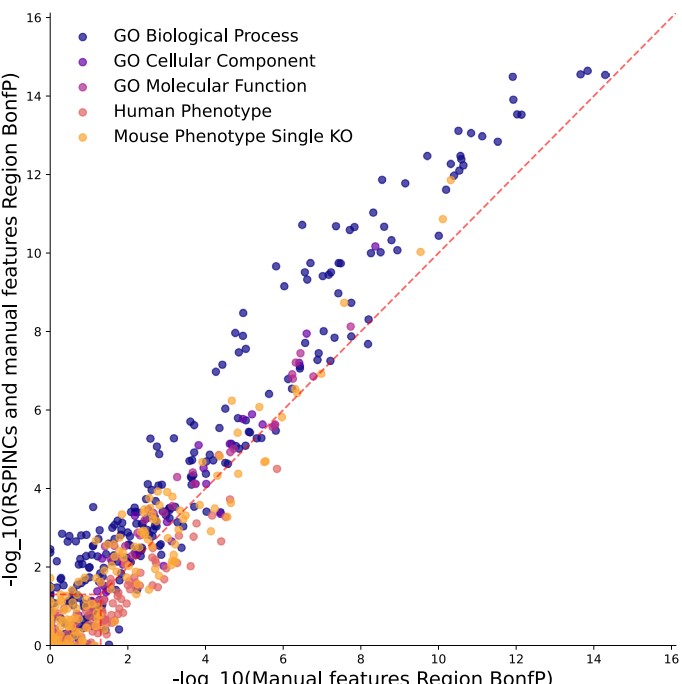

**Extended Data Fig. 8 | GREAT region-based enrichments for traditional measurements and RSPINCs.** The set of loci discovered through the union of traditional measurements and RSPINCs produces enrichments with lower P-values than the loci from traditional measurements alone. P-values were computed using the one-sided region-based binomial and Bonferroni-corrected for the number of tests performed.

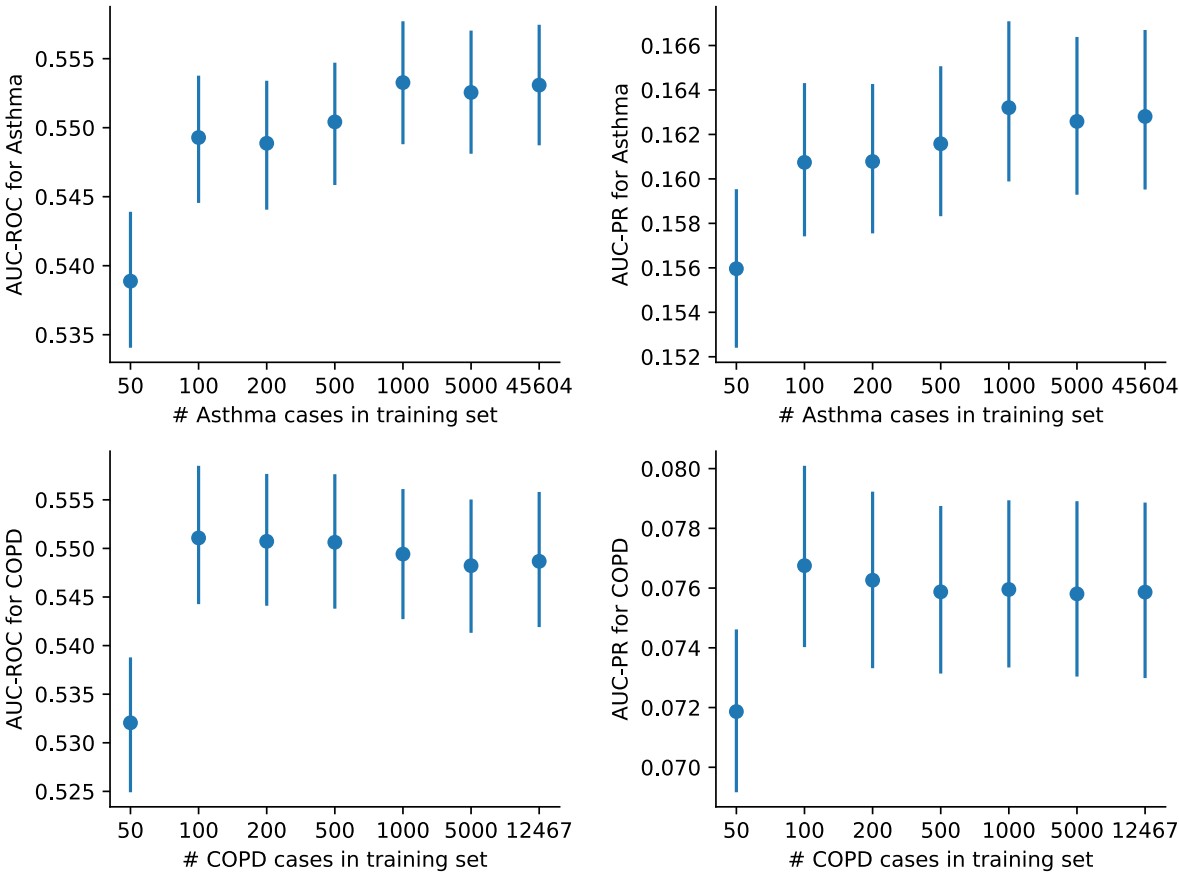

**Extended Data Fig. 9 | PRS performance under labeled training data ablation.** Datasets with balanced numbers of cases and controls were used to train PRS in European GIA individuals. In all figures, solid vertical intervals represent 95% confidence intervals generated by statistical bootstrapping (300 repetitions), and the center points are the bootstrapping means.

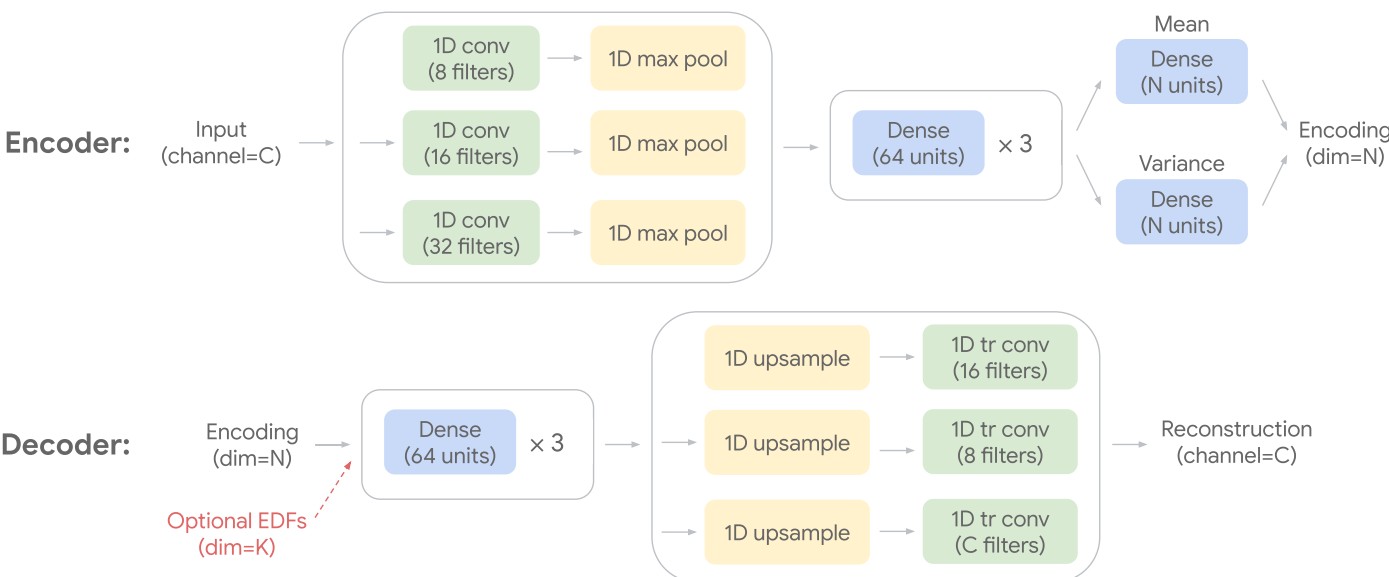

**Extended Data Fig. 10 | VAE model architecture details.** All convolution layers use the kernel size = 10, stride = 1, padding = 'same' and have the bias term. All max pooling and upsampling layers have a size of 2. All layers use the nonlinear ReLU activation, except for the 'mean' and 'variance' dense layers in the encoder. $N$ is the latent dimension (5 for SPINCs and PLENCs, 2 for RSPINCs and RPLENCs). $K$ is the number of EDFs (5 for both RSPINCs and RPLENCs). $C$ is the number of channels in the input, which is 2 when we encode both volume–time and flow–time curves together and 1 when we encode a single curve. Conv = convolution; tr conv = transposed convolution.

# Reporting Summary

## Statistics

For all statistical analyses, confirm that the following items are present in the figure legend, table legend, main text, or Methods section.

| n/a | Confirmed | |
|---|---|---|
| ☐ | ☒ | The exact sample size (*n*) for each experimental group/condition, given as a discrete number and unit of measurement |
| ☐ | ☒ | A statement on whether measurements were taken from distinct samples or whether the same sample was measured repeatedly |
| ☐ | ☒ | The statistical test(s) used AND whether they are one- or two-sided *Only common tests should be described solely by name; describe more complex techniques in the Methods section.* |
| ☐ | ☒ | A description of all covariates tested |
| ☐ | ☒ | A description of any assumptions or corrections, such as tests of normality and adjustment for multiple comparisons |
| ☐ | ☒ | A full description of the statistical parameters including central tendency (e.g. means) or other basic estimates (e.g. regression coefficient) AND variation (e.g. standard deviation) or associated estimates of uncertainty (e.g. confidence intervals) |
| ☐ | ☒ | For null hypothesis testing, the test statistic (e.g. *F*, *t*, *r*) with confidence intervals, effect sizes, degrees of freedom and *P* value noted *Give P values as exact values whenever suitable.* |
| ☒ | ☐ | For Bayesian analysis, information on the choice of priors and Markov chain Monte Carlo settings |
| ☒ | ☐ | For hierarchical and complex designs, identification of the appropriate level for tests and full reporting of outcomes |
| ☒ | ☐ | Estimates of effect sizes (e.g. Cohen's *d*, Pearson's *r*), indicating how they were calculated |

*Our web collection on statistics for biologists contains articles on many of the points above.*

## Software and code

Policy information about availability of computer code

| Data collection | No software is used for data collection. We have used available UK Biobank and COPDGene data. |
|---|---|

| Data analysis | Baseline and BaselineLD annotations: https://data.broadinstitute.org/alkesgroup/ldscore<br>BOLT-LMM (v2.3.6) software: https://data.broadinstitute.org/alkesgroup/bolt-lmm<br>COPDGene study: https://www.ncbi.nlm.nih.gov/projects/gap/cgi-bin/study.cgi?study_id=phs000179.v6.p2<br>GCTA (1.93.3beta) software: https://github.com/jianyangqt/gcta<br>GREAT (v4.0.4) software: http://great.stanford.edu<br>GWAS Catalog: https://www.ebi.ac.uk/gwas/<br>Indiana Biobank study: https://indianabiobank.org/<br>Pan-UK Biobank GWAS: https://pan.ukbb.broadinstitute.org<br>PLINK (v1.9) software: https://www.cog-genomics.org/plink1.9<br>TensorFlow: https://www.tensorflow.org<br>UCSC LiftOver: https://genome.ucsc.edu/cgi-bin/hgLiftOver<br>UK Biobank study: https://www.ukbiobank.ac.uk<br>Michigan Imputation Server https://imputationserver.sph.umich.edu/index.html#!pages/home<br>scikit-learn (v1.0.2): https://github.com/scikit-learn/scikit-learn<br>SciPy (v1.9.3): https://github.com/scipy/scipy |
|---|---|

For manuscripts utilizing custom algorithms or software that are central to the research but not yet described in published literature, software must be made available to editors and reviewers. We strongly encourage code deposition in a community repository (e.g. GitHub). See the Nature Portfolio guidelines for submitting code & software for further information.

## Data

Policy information about availability of data

All manuscripts must include a data availability statement. This statement should provide the following information, where applicable:
- Accession codes, unique identifiers, or web links for publicly available datasets
- A description of any restrictions on data availability
- For clinical datasets or third party data, please ensure that the statement adheres to our policy

UK Biobank study: https://www.ukbiobank.ac.uk and our access was approved under Application 65275. This research used data generated by the COPDGene study (dbGaP accession phs000179.v6.p2), which was supported by NIH grants U01 HL089856 and U01 HL089897. The EPIC-Norfolk study (DOI 10.22025/2019.10.105.00004) and Indiana Biobank. This research used data generated by the eMERGE III study which was obtained from dbGaP under accession phs001584.v2.p2. GWAS summary statistics for SPINCs, RSPINCs, PLENCs, RPLENCs, and EDFs are freely available on Google Cloud Storage at https://console.cloud.google.com/storage/browser/brain-genomics-public/research/regle.

## Research involving human participants, their data, or biological material

Policy information about studies with human participants or human data. See also policy information about sex, gender (identity/presentation), and sexual orientation and race, ethnicity and racism.

| Reporting on sex and gender | We did not collect new data for this work. We utilized the sex provided by UK Biobank. |
|---|---|
| Reporting on race, ethnicity, or other socially relevant groupings | We did not collect new data for this work. We utilized genetically inferred ancestry or reported in the study. UK Bionbank: We used individuals whose genetically inferred ancestry (GIA) is European, COPDGene: "Non-Hispanic White" subset (n=6,576) based on study reported ancestry and "African American" subset (n=3,140), eMERGE III: "White" subset (n=8,288) based on study reported ancestry, EPIC-Norfolk: "White" subset (n=8,288) based on study reported ancestry, and Indiana Biobank: All European GIA.We did not report any race or ethnicity. We did apply our method to different ethnicities to illustrate the capability of our method. |
| Population characteristics | We did not collect new data for this work. We utilize Age, genotype array, BMI, smoking status that are provided by UK Biobank data for GWAS. For COPDGene, eMERGE III, EPIC-Norfolk, and Indiana Biobank, we evaluate PRS on all samples. |
| Recruitment | We use UK Biobank sample and we did not perform any recruitment. |
| Ethics oversight | This study utilize public available data such as UK Biobank. |

Note that full information on the approval of the study protocol must also be provided in the manuscript.

# Field-specific reporting

Please select the one below that is the best fit for your research. If you are not sure, read the appropriate sections before making your selection.

☒ Life sciences ☐ Behavioural & social sciences ☐ Ecological, evolutionary & environmental sciences

For a reference copy of the document with all sections, see nature.com/documents/nr-reporting-summary-flat.pdf

# Life sciences study design

All studies must disclose on these points even when the disclosure is negative.

| | |
|---|---|
| Sample size | We use all the samples provided by UK Biobank. |
| Data exclusions | We removed samples with excess heterozygosity, missingness, or putative sex chromosome aneuploidy as defined by Bycroft et al 2018 (https://www.nature.com/articles/s41586-018-0579-z). We further limited to individuals of European genetic ancestry as defined in Alipanahi et al 2021 (https://doi.org/10.1016/j.ajhg.2021.05.004) |
| Replication | We have used GWAS catalog, previous GWAS (Shrine et al. 2023 Nature Genetics; DOI: https://doi.org/10.1038/s41588-018-0321-7 , Sakornsakolpat et al 2019 Nature Genetics; DOI: 10.1038/s41588-018-0342-2) to define nove loci. |
| Randomization | We use all the samples provided by UK Biobank. |
| Blinding | No blinding has been performed in our analysis. We use all the samples provided by UK Biobank. |

# Reporting for specific materials, systems and methods

We require information from authors about some types of materials, experimental systems and methods used in many studies. Here, indicate whether each material, system or method listed is relevant to your study. If you are not sure if a list item applies to your research, read the appropriate section before selecting a response.

## Materials & experimental systems

| n/a | Involved in the study |
|---|---|
| ☒ | ☐ Antibodies |
| ☒ | ☐ Eukaryotic cell lines |
| ☒ | ☐ Palaeontology and archaeology |
| ☒ | ☐ Animals and other organisms |
| ☒ | ☐ Clinical data |
| ☒ | ☐ Dual use research of concern |
| ☒ | ☐ Plants |

## Methods

| n/a | Involved in the study |
|---|---|
| ☒ | ☐ ChIP-seq |
| ☒ | ☐ Flow cytometry |
| ☒ | ☐ MRI-based neuroimaging |

