## [Peer Review File · Nature Genetics]

Peer Review Information

Manuscript Title: Unsupervised representation learning improves genomic discovery and disease risk prediction

Corresponding author name(s): Dr Farhad Hormozdiari, Dr Ted Yun, Dr Cory (Y) McLean

Reviewer Comments & Decisions:

Decision Letter, initial version:

1st Nov 2023

Dear Farhad,

Firstly, I would like to extend our apologies for the extended time in review to you and your co-authors, and thank you for your patience.

Your Article, "Unsupervised representation learning improves genomic discovery for lung function and respiratory disease prediction" has now been seen by 3 referees. You will see from their comments below that while they find your work of interest, some important points are raised. We are interested in the possibility of publishing your study in Nature Genetics, but would like to consider your response to these concerns in the form of a revised manuscript before we make a final decision on publication.

The referees think that REGLE has potential, saying this is an important contribution to the field, but also seem to be of the mind that this overall advance needs to be better-demonstrated.

Reviewer #1 is the most positive; they ask a number of questions on how to use REGLE, but these requests seem mostly minor.

Referee #2 also appreciates the potential but is more on the fence. The major thrust of their review asks the question: what is the value-add of using this VAE over simpler linear methods (i.e. PCA), which they do not think has been adequately demonstrated. They do, however, make concrete and constructive suggestions to address their concerns.

Reviewer #3 strikes a similar tone to #2, albeit with more positivity; again, they ask some important questions, including better justification for the use of the VAE.

In our reading of these reviews, the requests made do not seem onerous and we hope you will be able to address all of them. We'd particularly highlight the overlapping and fundamental concern of Referees #2 and #3: it is clear that the advance, and distinctiveness, of using REGLE's VAE approach over other methodologies must be more

strongly demonstrated.

To guide the scope of the revisions, the editors discuss the referee reports in detail within the team, including with the chief editor, with a view to identifying key priorities that should be addressed in revision and sometimes overruling referee requests that are deemed beyond the scope of the current study. We hope that you will find the prioritized set of referee points to be useful when revising your study. Please do not hesitate to get in touch if you would like to discuss these issues further.

We therefore invite you to revise your manuscript taking into account all reviewer and editor comments. Please highlight all changes in the manuscript text file. At this stage we will need you to upload a copy of the manuscript in MS Word .docx or similar editable format.

*2) If you have not done so already please begin to revise your manuscript so that it conforms to our Article format instructions, available here.

*3) Include a revised version of any required Reporting Summary: <https://www.nature.com/documents/nr-reporting-summary.pdf>

Please be aware of our guidelines on digital image standards.

[redacted]

Nature Genetics is committed to improving transparency in authorship. As part of our efforts in this direction, we are now requesting that all authors identified as 'corresponding author' on published papers create and link their Open Researcher and Contributor Identifier (ORCID) with their account on the Manuscript Tracking System (MTS),

prior to acceptance. ORCID helps the scientific community achieve unambiguous attribution of all scholarly contributions. You can create and link your ORCID from the home page of the MTS by clicking on 'Modify my Springer Nature account'. For more information please visit www.springernature.com/orcid.

Sincerely,

Michael Fletcher, PhD
Senior Editor, Nature Genetics

ORCID: 0000-0003-1589-7087

Referee expertise:

Referee #1: machine learning; cardiovascular genetics.

Referee #2: machine learning, human genetics.

Referee #3: lung genetics.

Reviewers' Comments:

Reviewer #1:

Remarks to the Author:

The authors describe a general approach to compress biological signal data into a latent representation, optionally residualized for known measurements, that is conducive to downstream analyses such as genetic analysis. Specifically, they show that spirometry and PPGs can be compressed into a very small number of dimensions, and that those latent representations embed biological signals yielding clinical and genetic associations.

The introduction sets the stage well. In particular, when I saw the title of this manuscript I thought of the work of Verweij [Cell Systems 2020] and I'm glad to see the Authors cite that work—it can be seen as the simplest approach to GWAS of signal data by acting as an identity encoding of the median waveform data. The Authors explain strengths and limitations of that approach and then march through prior work that has compressed biological signal data into more compact and less redundant representations, including some of the Authors' own recent publications, and the limitations of those approaches. The authors then describe their approach in the present manuscript, which uses a variational autoencoder-based approach that they term REGLE. Arguably there is other prior work using VAEs to represent biological data (albeit not for GWAS) that could be worth citing; for example, Gomari et al (Communications Biology 2022; PMID 35773471) used VAEs to learn latent representations of metabolomics data.

My view is that the key innovation in REGLE is the optional ability to learn a residual representation that is (largely) not already accounted for by clinically or expert defined features. This approach is not intrinsically tied to genetic analysis, but it assists with dimensionality reduction in a manner that may naturally facilitate genetic analyses.

One of the manipulable parameters when choosing a latent encoding is the dimension of the latent space. Here, for

residual spirogram encodings, the Authors demonstrate a scree-like plot in Supp Figure 2 showing reconstruction error as a function of latent dimension in their residual spirogram encoding model. Two points here, one about finding information and another about modeling choices. First, the Supp Figure 2 residual spirogram encoding plot begins with a latent dimension of zero, yielding the reconstruction error from using only the five classic spirogram readouts. This is a very helpful baseline but it would be nice to also be given a sense for a baseline without the five classical readouts already contributing. I believe that this information is provided in Supplementary Table 2, except without a $\text{dim}=0$ representation, so it might be resolved by referencing Supp Table 2 in the Sup Figure 2 caption. Second, the decision to retain two of these residual latent dimensions for the model, rather than one or three or six, does not seem to be stated. How was $\text{dim}=2$ chosen for the additional residual spirogram encodings? Even if it was chosen based on visual inspection, that should be stated. From my perspective, having to choose this dimension ends up introducing a degree of freedom into the analysis that is unavoidable but undesirable, and this is a key limitation of non-expert-based approaches to capturing information from signal data, even if a minor limitation. (In contrast, the rationale for the number of dimensions for the non-residual encodings [five] is clearly stated in the Methods as being the same number as the classical phenotypes.)

It would be helpful to have a more clear explanation of the model architecture. Methods 4.3 points to Supplement- >"SPINCs model architecture". This section does show a partial printout of the model architecture in a channels-last format. But the conv1d kernel size, as well as the stride, padding, and padding mode (if any) do not appear to be stated. Perhaps these could be inferred from the channels and parameter count, but this is not something I can do in my head as a reader. Allowing for that limitation, the display here is very helpful to understand the distinction between the spirogram encoding and the residual spirogram encoding models.

The model training procedure's optimizer, learning rate, and batch size are clearly stated in the supplement. Further, there is a clear demonstration that the output is consistent across random initializations. However, I did not see a statement about the number of epochs of training. Perhaps single-epoch training was performed? It would be helpful to have a little more detail to clarify the training procedure such as number of epochs, whether or not there was a learning rate scheduler in addition to the optimizer, whether early stopping based on the validation set was used, etc.

The model is trained and then applied within the same samples. In some contexts where generalization is a primary goal, that could be a problem (overfitting). From intuition, I don't think this is a particular problem for the present analytic setup, in part because the final latent dimensions chosen are vanishingly small. I suppose it's possible that training two models (one in half of the samples, the other in the other half) and applying them to the alternative half could reduce overfitting, if there is any. But again from the problem as formulated, this seems unlikely to be much of a problem. Raising it for the Authors' consideration, in case they think that overfitting is a topic that may be worth addressing in a sentence.

One potential limitation is that minimizing the VAE loss doesn't necessarily guarantee that these latent space encodings are conditioned to have the "best" biological meaning (from a subjective human perspective). Supp Figure 5 gives a pretty clear example of this - spirogram encoding 3 appears to represent nearly the same biological signal as FEV1 and FVC, while spirogram encoding 2 represents a partial recovery of the inverse of FEV1/FVC. However, despite those similarities, if we then look at Supp Table 9, the expert-defined value of FEV1 has a stronger hazard ratio and P value in the survival analysis than any of the VAE-learned spirogram encodings, even the most FEV1-like encoding #3. I think that this is a real limitation of the approach: where possible, smart people have probably come up with a more useful representation of the data than the VAE will. Nevertheless, it is due to this limitation that I also think that the Authors' residualization-based encodings are of the greatest value in this work, because these serve to salvage residual biological signals (shown in both PheWAS and GWAS in the main results) that would otherwise be discarded if we only used the expert-derived readouts. Further, the combined

polygenic information content of the VAE-learned representations surpasses that of the classical measurements in the Author's downstream genetic analyses, which is a departure from the phenotypic results and represents another strength of this approach.

Coming to the genetic analysis, analyzing only individuals with genetic identities similar to those of Europeans has been common due to concerns of population stratification, but the authors use a linear mixed modeling approach and incorporate PCs, which in many cases may mitigate stratification to a fair (albeit incomplete) extent. This is addressed in several places in the text, including the limitations. Modifying this step of the paper would entail a substantial amount of downstream computation, so I would just encourage the Authors to consider alternative defaults with respect to inclusion criteria in future work. The GWAS QC seems sensible.

In the GWAS results, the authors offer a nice comparison of loci to an extensive set of GWAS findings from prior literature. But the analysis that I really wanted to see was a head-to-head within-sample comparison of the EDFs with the spirometry encodings from the same samples.

I am a bit unclear on the polygenic score application. Line 293 says that these were computed in BOLT, but my understanding is that BOLT computes the score weights. Was it also used to apply the weights into individual-level scores, or was another tool used for that? The subsequent disease-specific PRS linear combination procedure is clear, and the ablation study shows that these disease-specific reweighting values could be learned from a very small portion of the training data if needed. The replication of the EDF vs spirogram encoding PRS across multiple cohorts is a strength.

I did not see a statement about human subjects research approvals or exemptions for most of the cohorts used in this analysis in the main text or the supplement.

Minor points:

- * I found the novel acronyms to be an impediment to reading the manuscript and its supplement (e.g., SPINC instead of "spirogram encoding"). However, the advantage of the acronyms is that there are clearly distinct words that distinguish "encoding" vs "residual encoding"—since these are critically different in terms of what they encode. Still, it would be nice if these terms did not have to be specific for each different input data type (SPINC/PLENC). I don't have a specific desired outcome here but wanted to mention it.

- * On line 443, the Authors mention, "First, by construction, the coordinates of the latent representation are minimally correlated [...]". I am very open to being corrected here, but my understanding that VAE latent dimensions are often uncorrelated, but not necessarily by construction—and if orthogonality is desired, then an explicit orthogonality loss can be imposed during training time which is indeed noted by the Authors in the Discussion. I agree that it is the case here that the latent representation yielded minimally correlated values as can be seen in the supplementary data, I am just not certain that this is true "by construction."

- * The layout of Supp Figure 7 emphasizes the time dimension while compressing the vertical axis, but the interesting signal has to do with the separation and sequence of the curves - would consider modifying the plots so that they are less compressed vertically.

- * Many heritable covariates are incorporated into the GWAS, including height, height², BMI, smoking status, and pack years of smoking. The motivation is clear. However, this might limit downstream use, because adjustment for heritable covariates can bias effect estimates (Aschard et al PMID 25640676). This could, in principle, influence some of the causal analyses as well. In principle, could consider a sensitivity analysis without these heritable covariates, although I also acknowledge that the specific summary statistics produced here are not the critical point of this paper and per the Code Availability section, the pretrained models should allow any investigator with UK Biobank access to run modified GWAS with different adjustments or inclusion criteria at will.

The simplicity of the Authors' approach, and its ability to account for expert-defined features, are two key strengths that make this an appealing tool for reference in future analyses.

Signed,

James Pirruccello

Reviewer #2:

Remarks to the Author:

Yun et al. present a method, based on convolutional VAEs, to learn representations of higher-dimensional clinical data in an unsupervised manner. They apply their method, REGLE, to two types of data: spirometry and photoplethysmograms (PPGs) and show that the resulting learned representations encode additional genetic information that is not captured by clinical biomarkers (termed expert-defined features or EDFs). In addition, they show that the PGS computed on the learned representations tend to be more predictive of relevant phenotypes and disease (asthma and COPD for spirometry; hypertension and systolic blood pressure for PPGs) and that these representations lead to the discovery of genetic variants not found with EDFs. Overall, this approach holds promise in exploring the diversity of phenotypes being collected in Biobanks. There are several comments that would be important to address:

Major comments:

1. a) It is described that 5 SPINCs and 5 PLENCs were learned to roughly match the number of clinical biomarkers in each domain. It appears in the range of 4~6 latent dimensions that both VAE and PCA reconstruct the data very well (Figures 2 and 3). What is the justification for using latent representations from a non-linear model when the linear model also appears sufficient for a similar number of dimensions?
1. b) For fewer dimensions REGLE does appear more statistically efficient than PCA but is this always desirable? I think it is important and valuable to understand in depth the gain in the VAE over a linear dimensionality reduction approach such as PCA (given the simplicity, stability, and the disentanglement due to the orthogonality of the PCs) especially given the PCs are quite comparable to VAE in terms of reconstruction for a comparable latent dimensionality. So a question for the authors is how would the PCs compare in the downstream tasks such as intermediate PGS or GWAS? How disentangled are the representations from the two methods in downstream analysis ?
2. As the precision recall (PR) would be important for Asthma and COPD and the PR between EDFs and SPINCs appear quite close, it would be good to state the standard errors and/or the p-values. Is this not more a sign that clinical annotations are very adequate?
3. The inclusion of EDFs in the latent dimension can be considered a novel analytical approach, but I would be hesitant to present it as a novel VAE architecture. It appears to be a conditional VAE which has been widely used to build VAEs that take covariates into account.
4. An established clinical biomarker may appear correlated with SPINC1 and again with SPINC3 (as shown in GWAS plots where MAPT and SCAPER reappear). While it is stated that VAEs can discover orthogonal latent features, is this true in practice? A convincing experiment would be to show the specificity of hits between the latent features and more generally to look at the polygenic architecture (by estimating genetic correlations). And if this is also true or not true for EDFs.

5. The comparisons of the GWAS hits SPINCs to Shrine et al. and the NHGRI GWAS catalog is interesting but there is a concern that the Shrine et al. study overlaps UKBB as would the GWAS catalog (unless the authors are excluding this).

6. How correlated are the learned representations with each other (eg. do signals re-appear in more than one latent factor)?

7. It is discussed that there are no guarantees in the direction (sign) of the learned latent features. While it is true we can re-orient eg. GWAS effects for relatively known phenotypes, would this not be a source of ambiguity for novel phenotypes?

Minor comments:

1. What is the scale of the reconstruction error? (e. g. vol^2)

2. The VAE learns both the mean and variance. Is the variance information used at all in this study?

Reviewer #3:

Remarks to the Author:

The authors present a method to improve phenotyping in genome-wide association studies that uses unsupervised deep representation learning and show that this method reveals structure in multidimensional data that is missed by existing expert-defined features. They show that this uncaptured structure implicates new associated genetic loci that have been missed in previous genetic association studies of lung and cardiovascular data. This is an important contribution to the field and paves the way for more in-depth phenotyping leading to greater capture of genetic effects for traits investigated by multidimensional data.

The method is very clearly described and the results clearly demonstrate the advantage of the method to extract information from high-dimensional clinical data in the two scenarios demonstrated in order to find new genetic loci and improve the accuracy of risk scores. The interpretation of RSPINCs is also interesting on how different RSPINCs affect different parts of the blow curve.

I have a few queries:

The use of VAE seems unnecessarily complex, couldn't the spirometry/PPGs be adequately modelled by a simple function with a similar number of parameters to a SPINC/PLENC (the examples in refs. 1-5 are more complex HDCC where deep learning would seem more necessary). It's not clear to me how the introduction of stochasticity and forcing of disentanglement helps modelling the relatively simple form of spirometry/PPGs. The fact that the spirometry can be reconstructed with just 5 SPINCs makes me wonder if they could equally be accurately modelled with a cubic spline with 5 degrees of freedom?

Why 80/20 training/validation sets?

Why is there no investigation of EDF + RPLENCs for PPGs?

Line 247-248 - how does the hazard per SD of SPINC/PLENC compare to hazard per SD FEV1 or FVC?

Line 262: How does heritability of (R)SPINCs/PLENCs compare to heritability of EDFs?

Line 272-273: Shrine et al. used $5E-9$, you are using $5E-8$, shouldn't you compare number of loci found and same level of significance?

I don't see how the GARFIELD analysis demonstrates any improvement over EDFs?

I don't see what the causality analysis adds, aren't COPD and hypertension just defined by thresholds of spirometry and PPGs respectively?

Line 72-74: 2 and 3 are the same issue, redundant multiple testing

Line 256-257: could just say Supplementary Figures 12-23

Between Supplementary Table 12 & 13 the captions have switched from being above the table to below it.

Would Supplementary Tables 14-17 suggest it is better to use SPINCs than EDF+RSPINCs?

Author Rebuttal to Initial comments

Response to reviewers for NG-A62621R1

Reviewers' Comments:

Reviewer #1

Remarks to the Author:

The authors describe a general approach to compress biological signal data into a latent representation, optionally residualized for known measurements, that is conducive to downstream analyses such as genetic analysis.

Specifically, they show that spirograms and PPGs can be compressed into a very small number of dimensions, and that those latent representations embed biological signals yielding clinical and genetic associations.

The introduction sets the stage well. In particular, when I saw the title of this manuscript I thought of the work of Verweij [Cell Systems 2020] and I'm glad to see the Authors cite that work—it can be seen as the simplest approach to GWAS of signal data by acting as an identity encoding of the median waveform data. The Authors explain strengths and limitations of that approach and then march through prior work that has compressed biological signal data into more compact and less redundant representations, including some of the Authors' own recent publications, and the limitations of those approaches. The authors then describe their approach in the present manuscript, which uses a variational autoencoder-based approach that they term REGLE. Arguably there is other prior work using VAEs to represent biological data (albeit not for GWAS) that could be worth citing; for example, Gomari et al (Communications Biology 2022; PMID 35773471) used VAEs to learn latent representations of metabolomics data.

We thank the reviewer for the positive feedback. We added the Gomari et al (Communications Biology 2022; PMID 35773471) citation to introduce VAE methods. We update the introduction (page 5):

“REGLE is based on the variational autoencoder (VAE) [29] model. Although VAEs have previously been applied to metabolomics data [30], the utility of VAE embeddings for GWAS, polygenic risk scores (PRSs), and downstream analyses have not been previously explored.”

My view is that the key innovation in REGLE is the optional ability to learn a residual representation that is (largely) not already accounted for by clinically or expert defined features. This approach is not intrinsically tied to genetic analysis, but it assists with dimensionality reduction in a manner that may naturally facilitate genetic analyses.

We agree with the Reviewer's assessment. To emphasize this innovation, while simultaneously addressing Reviewer 2's Comment #3 about previously known models such as conditional VAEs, we have modified the Discussion text as follows (p 19):

Prior text: "The architecture modification we introduce in REGLE to support expert-defined features (EDFs) enables a principled use of expert human knowledge and encourages the remaining latent coordinates to encode biological function explicitly not captured by the EDFs."

Current text: "To support the principled use of expert-defined features (EDFs) in modeling, REGLE uses a modification of VAE in which the EDFs are additionally included in the input to the decoder, implicitly encouraging the encoder to learn features not captured by the EDFs. Conceptually, this is closely related to conditional VAE (Sohn et al 2015), a conditional generative model created to enable class-conditioned sample generation. The main difference between our approach and conditional VAE is that our encoder neither takes EDFs as input nor uses a conditional prior network. As a result, our trained encoder can generate residual encodings directly from input HDCD without using EDFs as an additional input."

We also add a point to limitations/future work to the Discussion to highlight that the existing REGLE objective function for residual representation does not directly maximize genetic signal: "Third, the VAE objective and in particular the reconstruction error is not necessarily optimal for genetic analyses, and explicitly incorporating an objective to maximize heritability of the learned representation may be a fruitful line of future research to maximize genetic discovery through unsupervised learning (Zhou et al 2015)."

Finally, to demonstrate the generalizability of the residual approach, we compute and add "RPLENCs" (residual PPG encodings) results to Table 1 as well.

One of the manipulable parameters when choosing a latent encoding is the dimension of the latent space. Here, for residual spirogram encodings, the Authors demonstrate a scree-like plot in Supp Figure 2 showing reconstruction error as a function of latent dimension in their residual spirogram encoding model. Two points here, one about finding information and another about modeling choices. First, the Supp Figure 2 residual spirogram encoding plot begins with a latent dimension of zero, yielding the reconstruction error from using only the five classic spirogram readouts. This is a very helpful baseline but it would be nice to also be given a sense for a baseline without the five classical readouts already contributing. I believe that this information is provided in Supplementary Table 2, except without a dim=0 representation, so it might be resolved by referencing Supp Table 2 in the Sup Figure 2 caption.

We thank the reviewer for the suggestion. To calculate a baseline error without the five EDFs with the latent dimension of zero (in which case we do not have any individual-level coordinates to reconstruct the curves from), we computed the average spirogram in the training set and evaluated the reconstruction error using the average spirogram as the reconstruction for all individuals. This will give a sense of how well the models capture the individual deviations from the average spirogram in a population. We added this information in the legend of Supplementary Figure 2:

"For comparison, using the training set average curve for all evaluation set individuals would yield the reconstruction error of 2.057727, 20× higher than any data point in this figure."

We also added a new row in Supplementary Table 2 to include the reconstruction error from RSPINCs with the latent dimension of zero (i.e. using just the five EDFs).

Second, the decision to retain two of these residual latent dimensions for the model, rather than one or three or six, does not seem to be stated. How was dim=2 chosen for the additional residual spirogram encodings? Even if it was chosen based on visual inspection, that should be stated. From my perspective, having to choose this dimension ends up introducing a degree of freedom into the analysis that is unavoidable but undesirable, and this is a key limitation of

non-expert-based approaches to capturing information from signal data, even if a minor limitation. (In contrast, the rationale for the number of dimensions for the non-residual encodings [five] is clearly stated in the Methods as being the same number as the classical phenotypes.)

We have modified the Methods section 4.3 (p 24) to address this question:

"We chose to use two latent dimensions for the encoder based on REGLE strong reconstruction performance (Supplementary Figure 2) while maintaining a comparable number of total latent dimensions to SPINCs."

It would be helpful to have a more clear explanation of the model architecture. Methods 4.3 points to Supplementary Figure 2 "SPINCs model architecture". This section does show a partial printout of the model architecture in a channels-last format. But the conv1d kernel size, as well as the stride, padding, and padding mode (if any) do not appear to be stated. Perhaps these could be inferred from the channels and parameter count, but this is not something I can do in my head as a reader. Allowing for that limitation, the display here is very helpful to understand the distinction between the spirogram encoding and the residual spirogram encoding models.

We thank the reviewer for pointing this out. We added Supplementary Figure 44 describing more details about the model such as convolution kernel size, stride, padding, activation, etc., and added references to the new supplementary figure in the "Convolutional VAE model architecture and training" section in Methods.

We'd also like to note that the complete model definition written in TensorFlow/Keras is publicly available in our open-source code repository (see lib/models.py): <https://github.com/Google-Health/genomics-research/tree/main/regle>

The model training procedure's optimizer, learning rate, and batch size are clearly stated in the supplement. Further, there is a clear demonstration that the output is consistent across random initializations. However, I did not see a statement about the number of epochs of training.

Perhaps single-epoch training was performed? It would be helpful to have a little more detail to clarify the training procedure such as number of epochs, whether or not there was a learning rate scheduler in addition to the optimizer, whether early stopping based on the validation set was used, etc.

We thank the reviewer for pointing this out. We modified the following sentences in the "Convolutional VAE model architecture and training" section in Methods to include this information:

"For optimization, the Adam optimizer [65] is used with varying learning rates and batch sizes. No learning rate scheduler was used. After training for 100 epochs, the final learning rate and batch size values ("hyperparameters") for (R)SPINCs and PLENCs were chosen to minimize the VAE loss in the validation set (Supplementary Notes, Supplementary Table 1)."

The model is trained and then applied within the same samples. In some contexts where generalization is a primary goal, that could be a problem (overfitting). From intuition, I don't think this is a particular problem for the present analytic setup, in part because the final latent dimensions chosen are vanishingly small. I suppose it's possible that training two models (one in half of the samples, the other in the other half) and applying them to the alternative half could reduce overfitting, if there is any. But again from the problem as formulated, this seems unlikely to be much of a problem. Raising it for the Authors' consideration, in case they think that overfitting is a topic that may be worth addressing in a sentence.

We thank the reviewers for raising this issue. We performed multiple tests to make sure overfitting is not occurring in our training. First, we plot the MSE of the model for SPINCs for both train and validation datasets (Figure 1 response to the reviewer) and we observed a similar pattern between train and validation.

Figure 1. Reconstruction error of SPINCs between Train and Validation datasets.

Second, we performed a similar analysis for PLENCs and observed a similar pattern between train and validation reconstruction error (Figure 2).

Figure 2. Reconstruction error of PLENCs between Train and Validation datasets.

We believe the chance of overfitting is extremely low given both the observed results and the limited latent embedding size. Furthermore, even if overfitting occurred, it should not affect the downstream genetic analysis (GWAS) and, in the case of PRS, all our evaluations were performed in a separate test dataset never used for training the model.

However, the generalization of our model to newer dataset is a very interesting and important question. We add this as a possible limitation of the work (page 22, Discussion) which requires additional spirogram/PPG data to fully investigate the generalizability of REGLE embedding to spirogram/PPG.

“Seventh, REGLE was trained on spiograms and PPG obtained from UK Biobank only; thus, (R)SPINCs and (R)PLENCs representations may not generalize well to other datasets. One needs additional datasets to investigate the generalizability of the encodings.”

Lastly, regarding “possible that training two models (one in half of the samples, the other in the other half) and applying them to the alternative half could reduce overfitting”, we initially consider this procedure; however, given the fact that the coordinates of the two VAE embeddings may not necessary align (e.g. the first coordinate of the embedding from model 1 may correspond to the fifth coordinate of the embedding from model 2) as these embeddings can be any permutation of each other (Supplementary Figure 3), we did not use this approach for training the VAE.

One potential limitation is that minimizing the VAE loss doesn’t necessarily guarantee that these latent space encodings are conditioned to have the “best” biological meaning (from a subjective human perspective). Supp Figure 5 gives a pretty clear example of this - spirogram encoding 3 appears to represent nearly the same biological signal as FEV1 and FVC, while spirogram encoding 2 represents a partial recovery of the inverse of FEV1/FVC. However, despite those similarities, if we then look at Supp Table 9, the expert-defined value of FEV1 has a stronger hazard ratio and P value in the survival analysis than any of the VAE-learned spirogram encodings, even the most FEV1-like encoding #3.

I think that this is a real limitation of the approach: where possible, smart people have probably come up with a more useful representation of the data than the VAE will. Nevertheless, it is due to this limitation that I also think that the Authors' residualization-based encodings are of the greatest value in this work, because these serve to salvage residual biological signals (shown in both PheWAS and GWAS in the main results) that would otherwise be discarded if we only used the expert-derived readouts. Further, the combined polygenic information content of the VAE-learned representations surpasses that of the classical measurements in the Author's downstream genetic analyses, which is a departure from the phenotypic results and represents another strength of this approach.

We agree with the reviewer that the objective function of VAE and in particular reconstruction error is not necessarily optimal for genetic analyses. In fact, we are interested in lower dimensional representations that maximize genetic results (GWAS and PRS). We consider the new objective function outside of the scope of this work. We add this as a limitation of our work to Discussion (page 21):

“Third, the VAE objective and in particular the reconstruction error is not necessarily optimal for genetic analyses, and explicitly incorporating an objective to maximize heritability of the learned representation may be a fruitful line of future research to maximize genetic discovery through unsupervised learning (Zhou et al 2015).”

Coming to the genetic analysis, analyzing only individuals with genetic identities similar to those of Europeans has been common due to concerns of population stratification, but the authors use a linear mixed modeling approach and incorporate PCs, which in many cases may mitigate stratification to a fair (albeit incomplete) extent. This is addressed in several places in the text, including the limitations. Modifying this step of the paper would entail a substantial amount of downstream computation, so I would just encourage the Authors to consider alternative defaults with respect to inclusion criteria in future work. The GWAS QC seems sensible.

We agree with the reviewer that performing GWAS on non-European individuals is important for equitability and fairness considerations, and in other unpublished work we have now extended our analyses to support multi-ancestry GWAS. We also note that to make sure our SPINCs PRS is not biased to one population, we analyzed the African American samples in COPDGene (Supplementary Table 21) as part of PRS transferability.

In the GWAS results, the authors offer a nice comparison of loci to an extensive set of GWAS findings from prior literature. But the analysis that I really wanted to see was a head-to-head within-sample comparison of the EDFs with the spirometry encodings from the same samples.

The GWAS comparison performed on Table 1 of manuscript between REGLE (SPINCs and PLENCs) and EDFs are done on UK Biobank using the same set of individuals. To make this clear, the Results section wording now reads (p 11):

“We then performed GWAS on all European-ancestry individuals across all encoding coordinates (SPINCs, RSPINCs, PLENCs, and RPLENCs) and 5 EDFs using BOLT-LMM [38, 39], adjusting for covariates (Supplementary Notes; Methods; Supplementary Figures 13 to 26).”

I am a bit unclear on the polygenic score application. Line 293 says that these were computed in BOLT, but my understanding is that BOLT computes the score weights. Was it also used to apply the weights into individual-level scores, or was another tool used for that? The subsequent disease-specific PRS linear combination procedure is clear, and the ablation study shows that these disease-specific reweighting values could be learned from a very small portion of the training data if needed. The replication of the EDF vs spiogram encoding PRS across multiple cohorts is a strength.

$$s_{ik} = \sum_{j=1}^M g_{ij} \hat{\beta}_{jk}$$

Where g_{ij} is the i -th individual genotype at the j -th variant and M is the total number of variants or SNPs. $\hat{\beta}_{jk}$ is the effect size estimated by BOLT-LMM for j -th variant and k -th latent dimension. Next, we estimate the i -th individual phenotype of interest as follows:

$$y_i = \sum_{k=1}^T s_{jk} w_k$$

Where w_k is the linear effect size estimated via the linear model. In all cases that we have five latent embeddings we set T to 5.

We updated the Method section (pages 28-29) with the above description.

I did not see a statement about human subjects research approvals or exemptions for most of the cohorts used in this analysis in the main text or the supplement.

We thank the reviewer for bringing this to our attention. We added these statements to the “Dataset acknowledgment” section in Supplementary Notes.

“Advarra IRB (Columbia, MD) waived ethical approval for this work involving de-identified medical imagery and metadata under 45 CFR 46. Work related to genomics data were additionally reviewed by the respective data sources: UK Biobank, COPDGene, eMERGE III, EPIC Norfolk, and Indiana Biobank.”

Minor points:

* I found the novel acronyms to be an impediment to reading the manuscript and its supplement (e.g., SPINC instead of “spiogram encoding”). However, the advantage of the acronyms is that there are clearly distinct words that distinguish “encoding” vs “residual encoding”—since these are critically different in terms of what they encode. Still,

it would be nice if these terms did not have to be specific for each different input data type (SPINC/PLENC). I don't have a specific desired outcome here but wanted to mention it.

We thank the reviewer for the feedback and we acknowledge the readers would need some time to familiarize with the acronyms. As the reviewer also pointed out, we wanted to clearly distinguish the residual encodings from the non-residual counterparts, and also make it clear for which data modality the encodings were trained. We use REGLE for the general idea and methods applied throughout the paper.

* On line 443, the Authors mention, "First, by construction, the coordinates of the latent representation are minimally correlated [...]". I am very open to being corrected here, but my understanding that VAE latent dimension are often uncorrelated, but not necessarily by construction—and if orthogonality is desired, then an explicit orthogonality loss can be imposed during training time which is indeed noted by the Authors in the Discussion. I agree that it is the case here that the latent representation yielded minimally correlated values as can be seen in the supplementary data, I am just not certain that this is true "by construction."

We thank the reviewer for raising this point. The relative disentanglement of the VAE encodings is a result of using a factorized Gaussian prior distribution (and minimizing KL-divergence to that prior) that is included in the construction of VAE. More discussion of this topic can be found in [Burgess et al, 2017, arXiv:1804.03599], for example. We agree with the reviewer that adding an additional explicit orthogonality loss term for a measure of correlation or amplifying the KL-divergence term, as it is done in β -VAE [Higgins et al, ICLR 2017], FactorVAE [Kim & Mnih, PMLR 2018], β -TCVAE [Chen et al, NeurIPS 2018], etc., could be an interesting future direction, as we note in Discussion.

For additional clarity we modified the sentence in Discussion as follows:

"First, due to the usage of factorized Gaussian prior distribution in VAEs, the coordinates of the latent representation are minimally correlated, which increases the combined power of the downstream GWASs."

* The layout of Supp Figure 7 emphasizes the time dimension while compressing the vertical axis, but the interesting signal has to do with the separation and sequence of the curves - would consider modifying the plots so that they are less compressed vertically.

We thank the reviewer for the suggestion. We've updated each of our Kaplan-Meier figures (Supplementary Figures 7-10) to contain two columns so that each subfigure is compressed on the time axis rather than on the probability axis.

* Many heritable covariates are incorporated into the GWAS, including height, height², BMI, smoking status, and pack years of smoking. The motivation is clear. However, this might limit downstream use, because adjustment for heritable covariates can bias effect estimates (Aschard et al PMID 25640676). This could, in principle, influence some of the causal analyses as well. In principle, could consider a sensitivity analysis without these heritable covariates,

although I also acknowledge that the specific summary statistics produced here are not the critical point of this paper and per the Code Availability section, the pretrained models should allow any investigator with UK Biobank access to run modified GWAS with different adjustments or inclusion criteria at will.

We thank the reviewer for the suggestion. We performed GWAS without including the height, height², BMI, smoking status, and pack years of smoking for SPINCs/RSPINCs, and PLENCS. We observed that nearly all results are qualitatively equivalent to those with heritable covariates included, with only SPINC₃ associations with thyrotoxicosis and gluten-free diet no longer significant when excluding heritable covariates (Table 1 response to the reviewer). However, given that this is not the main analysis of the paper (comment from reviewer3: “I don't see what the causality analysis adds”), we have removed the causality analysis from the current version of the paper.

Trait1	Trait2	Include heritable covars.			Exclude heritable covars.		
		GCP (SE)	Log10 p	pg	GCP (SE)	Log10 p	pg
SPINC ₁	Asthma	-0.26(0.10)	-2.2	0.21(0.04)	-0.27(0.10)	-2.4	0.21(0.04)
SPINC ₂	Asthma	-0.19(0.12)	-0.9	0.41(0.04)	-0.18(0.11)	-0.9	0.38(0.04)
SPINC ₃	Asthma	-0.29(0.09)	-3.7	-0.22(0.05)	-0.46(0.15)	-4.5	-0.19(0.04)
SPINC ₄	Asthma	-0.03(0.10)	-0.1	-0.21(0.06)	-0.05(0.10)	-0.1	-0.21(0.06)
SPINC ₅	Asthma	-0.71(0.12)	-42.1	-0.20(0.06)	-0.75(0.12)	-44.9	-0.20(0.06)
SPINC ₁	COPD	0.12(0.15)	-0.3	0.31(0.07)	0.11(0.17)	-0.3	0.34(0.07)
SPINC ₂	COPD	0.82(0.14)	-6.7	0.52(0.08)	0.82(0.14)	-6.6	0.49(0.08)

SPINC ₃	COPD	0.47(0.33)	-0.5	-0.27(0.07)	0.04(0.39)	0	-0.29(0.07)
SPINC ₄	COPD	0.15(0.31)	-0.2	-0.07(0.08)	0.00(0.28)	0	-0.07(0.08)
SPINC ₅	COPD	0.07(0.55)	-0.2	0.08(0.10)	0.05(0.57)	-0.1	0.06(0.10)
SPINC ₁	Lupus	-0.10(0.52)	-0.1	-0.07(0.65)	-0.13(0.50)	-0.1	-0.06(0.54)
SPINC ₂	Lupus	-0.75(0.17)	-7.2	0.33(0.23)	-0.75(0.17)	-7	0.31(0.31)
SPINC ₃	Lupus	0.11(0.51)	-0.1	-0.25(0.17)	0.00(0.58)	-0.2	-0.15(0.10)
SPINC ₄	Lupus	-0.26(0.18)	-0.4	-0.05(1.10)	-0.41(0.26)	-0.5	-0.05(1.20)
SPINC ₅	Lupus	-0.02(0.59)	-0.2	-0.18(0.51)	-0.02(0.59)	-0.2	-0.16(0.56)
SPINC ₁	Sarcoidosis	-0.27(0.40)	-0.2	0.03(0.05)	-0.15(0.45)	-0.1	0.01(0.05)
SPINC ₂	Sarcoidosis	-0.14(0.27)	-0.3	0.20(0.23)	-0.18(0.25)	-0.4	0.18(0.21)
SPINC ₃	Sarcoidosis	-0.83(0.12)	-21.4	-0.24(0.10)	-0.66(0.23)	-8	-0.13(0.08)
SPINC ₄	Sarcoidosis	0.23(0.45)	-0.1	-0.19(0.12)	0.25(0.44)	-0.1	-0.20(0.12)
SPINC ₅	Sarcoidosis	-0.01(0.58)	-0.1	-0.06(0.22)	0.00(0.58)	-0.1	-0.05(0.21)
SPINC ₁	Thyrotoxicosis	-0.43(0.35)	-0.5	0.08(0.06)	-0.45(0.34)	-0.5	0.08(0.06)
SPINC ₂	Thyrotoxicosis	-0.08(0.61)	-0.5	0.16(0.22)	-0.39(0.42)	-0.6	0.14(0.20)
SPINC ₃	Thyrotoxicosis	-0.78(0.15)	-13.9	-0.24(0.10)	0.12(0.38)	-0.6	-0.14(0.09)
SPINC ₄	Thyrotoxicosis	-0.08(0.45)	-0.1	-0.01(0.19)	-0.03(0.53)	-0.1	-0.02(0.19)
SPINC ₅	Thyrotoxicosis	-0.43(0.35)	-0.6	-0.11(0.19)	-0.08(0.56)	-0.3	-0.09(0.19)
SPINC ₁	Gluten-free-diet	-0.59(0.25)	-2.2	0.17(0.09)	-0.59(0.25)	-2.2	0.17(0.10)
SPINC ₂	Gluten-free-diet	0.24(0.52)	-0.3	0.28(0.24)	0.36(0.42)	-0.4	0.25(0.21)
SPINC ₃	Gluten-free-diet	-0.68(0.22)	-4.7	-0.18(0.14)	-0.15(0.26)	-0.3	-0.09(0.14)
SPINC ₄	Gluten-free-diet	-0.35(0.33)	-0.2	-0.03(0.23)	-0.41(0.32)	-0.2	-0.04(0.24)
SPINC ₅	Gluten-free-diet	-0.02(0.58)	-0.2	-0.24(0.16)	-0.00(0.57)	-0.2	-0.21(0.16)
PLENC ₁	hypertension	-0.35(0.10)	-4.3	0.23(0.07)	-0.38(0.10)	-5.5	0.23(0.07)
PLENC ₂	hypertension	-0.05(0.15)	-0.1	-0.32(0.05)	-0.12(0.16)	-0.3	-0.32(0.05)
PLENC ₃	hypertension	-0.23(0.10)	-1.8	0.16(0.05)	-0.26(0.11)	-1.9	0.17(0.05)
PLENC ₄	hypertension	-0.35(0.27)	-0.5	0.40(0.10)	-0.41(0.25)	-0.8	0.39(0.09)
PLENC ₅	hypertension	-0.33(0.40)	-0.3	-0.09(0.05)	-0.33(0.43)	-0.4	-0.11(0.05)

Table 1. Compassion LCV results including and excluding heritable covariates. We compared the results of LCV when we included and excluded heritable covariates (e.g., height, BMI, smokers, etc) from GWAS.

The simplicity of the Authors' approach, and its ability to account for expert-defined features, are two key strengths that make this an appealing tool for reference in future analyses.

We thank the reviewer for the detailed and supportive feedback.

Signed,

James Pirruccello

Reviewer #2

Remarks to the Author:

Yun et al. present a method, based on convolutional VAEs, to learn representations of higher-dimensional clinical data in an unsupervised manner. They apply their method, REGLE, to two types of data: spiromograms and photoplethysmograms (PPGs) and show that the resulting learned representations encode additional genetic information that is not captured by clinical biomarkers (termed expert-defined features or EDFs). In addition, they show that the PGS computed on the learned representations tend to be more predictive of relevant phenotypes and disease (asthma and COPD for spiromograms; hypertension and systolic blood pressure for PPGs) and that these representations lead to the discovery of genetic variants not found with EDFs.

Overall, this approach holds promise in exploring the diversity of phenotypes being collected in Biobanks. There are several comments that would be important to address:

Major comments:

1. a) It is described that 5 SPINCs and 5 PLENCs were learned to roughly match the number of clinical biomarkers in each domain. It appears in the range of 4~6 latent dimensions that both VAE and PCA reconstruct the data very well (Figures 2 and 3). What is the justification for using latent representations from a non-linear model when the linear model also appears sufficient for a similar number of dimensions?

We thank the reviewer for raising this question. The FEV1/FVC ratio is a nonlinear function that has been used in GOLD definition of COPD and GWAS of FEV1/FVC (Shrine et al. 2023 Nature Genetics, Shrine et al. 2019 Nature Genetics, Hobbs et al. 2017 Nature Genetics), and it is one of the most important lung function indicators used in clinical settings. Thus, utilizing nonlinear representation is vital. Furthermore, one of the main advantages of REGLE is that it does not need the signal to always be nonlinear in cases where the best model is linear REGLE will find the linear representation as well.

Regarding using the 5 latent factors for SPINCs and PLENCs is to make sure our model comparisons are fair with respect to 5 commonly-used EDFs.

We made the following changes:

- 1) Result section "Overview of REGLE" (p 5): "REGLE consists of three main steps: 1) learning a non-linear and linear, low-dimensional, disentangled representation

..”

2) Introduction (p 4): “FEV1/FVC (non-linear function of FVC and FEV₁)”.

3) Discussion (p 18): “A key strength of our method is the use of a VAE to generate the low-dimensional (non-linear or linear) representations of HDCD. Considering the non-linear nature of biological networks and the significance of FEV1/FVC as a

valuable metric for assessing lung function is evident, particularly in clinical applications such as COPD diagnosis [17] and genetic analyses [33, 24, 25]. Therefore, incorporating non-linear capabilities is imperative in any models handling genetic data.)”

1. b) For fewer dimensions REGLE does appear more statistically efficient than PCA but is this always desirable? I think it is important and valuable to understand in depth the gain in the VAE over a linear dimensionality reduction approach such as PCA (given the simplicity, stability, and the disentanglement due to the orthogonality of the PCs) especially given the PCs are quite comparable to VAE in terms of reconstruction for a comparable latent dimensionality. So a question for the authors is how would the PCs compare in the downstream tasks such as intermediate PGS or GWAS? How disentangled are the representations from the two methods in downstream analysis ?

We compared REGLE and PCA results on spiromograms and PPG modalities using 3 different metrics: i) GWAS power via the expected chi-squared statistics ($E[\chi^2]$ Loh et al. 2015 Nature Genetics; Loh et al. 2018 Nature Genetics), ii) Number of loci/hits and iii) PRS prediction on various binary/continuous phenotypes. We observed that SPINCs and PLENCs have much higher expected Chi-squared statistics than PCA on various numbers of latent dimensions (Tables 2 and 3 respond to reviewer and Supplementary Tables 11-12 of the manuscript). In addition, Table 1 of the manuscript compared the number of loci/hits of different methods indicating SPINCs and PLENCs detect more previous known hits/loci and detect additional novel loci for lung and cardiovascular traits. Lastly, we compare SPINCs/PLENCs PRS with PCA on various phenotypes (Supplementary Table 18 for asthma, Supplementary Table 19 for COPD in UK Biobank, Supplementary Table 21 for COPD in COPDGene, Supplementary Table 23 for hypertension (HTN) in UK Biobank, and Supplementary Table 24 for systolic blood pressure (SBP) in UK Biobank).

We added the following changes to the main text of manuscript:

- Added expected chi-squared power comparison to result (page 11): “Furthermore, SPINCs and PLENCs GWAS have higher power compared to PCA method when expected chi-square statistics is used as measure of GWAS power \cite{Loh2015, Loh2018} (Supplementary Tables 11 and 12; Methods).”
- Added Supplementary Tables 11 and 12.
- Added a method section to illustrate the process of computing expected chi-square statistics for all methods.
- Added Supplementary Tables 23-24 to compare PRS between PLENCs and PCA for HTN and SBP.

PCA (5)	6.615 (0.002)	60.626 (1.208)
---------	---------------	----------------

PCA (4)	5.564 (0.002)	58.522 (1.207)
PCA (3)	4.470 (0.002)	55.503 (1.177)
PCA (2)	3.380 (0.002)	52.874 (1.168)
PCA (1)	1.861 (0.001)	16.269 (0.335)

Table 2. GWAS power compassion between SPINCs and PCA on Spirogram data.

Method	$E[\chi^2]$ All variants	$E[\chi^2]$ GWAS catalog Hits
PLENCs	5.651 (0.001)	15.805 (0.449)
PCA (5)	5.409 (0.001)	11.755 (0.329)
PCA (4)	4.354 (0.001)	10.109 (0.301)
PCA (3)	3.201 (0.0007)	6.159 (0.149)
PCA (2)	2.173 (0.0006)	4.692 (0.133)
PCA (1)	1.083 (0.0004)	1.831 (0.0659)

Table 3. GWAS power compassion between PLENCs and PCA on PPG data.

2. As the precision recall (PR) would be important for Asthma and COPD and the PR between EDFs and SPINCs appear quite close, it would be good to state the standard errors and/or the p-values. Is this not more a sign that clinical annotations are very adequate?

We added a 95% confidence interval to all estimates of all PRS results.

3. The inclusion of EDFs in the latent dimension can be considered a novel analytical approach, but I would be hesitant to present it as a novel VAE architecture. It appears to be a conditional VAE which has been widely used to build VAEs that take covariates into account.

We appreciate the reviewer’s concern, and have modified the manuscript to address it while highlighting the differences from prior conditional VAE. In the Discussion, we have made the following edit (also referenced in response to Reviewer 1’s comment about the “key innovation of REGLE”):

Prior text: “The architecture modification we introduce in REGLE to support expert-defined features (EDFs) enables a principled use of expert human knowledge and

encourages the remaining latent coordinates to encode biological function explicitly not captured by the EDFs.”

Current text: "To support the principled use of expert-defined features (EDFs) in modeling, REGLE uses a modification of VAE in which the EDFs are additionally included in the input to the decoder, implicitly encouraging the encoder to learn features not captured by the EDFs. Conceptually, this is closely related to conditional VAE (Sohn et al 2015), a conditional generative model created to enable class-conditioned sample generation. The main difference between our approach and conditional VAE is that our encoder neither takes EDFs as input nor uses a conditional prior network. As a result, our trained encoder can generate residual encodings directly from input HDCD without using EDFs as an additional input."

We have also replaced “novel VAE architecture” with “modified VAE architecture” in the Methods section 4.3 to address the reviewer's concern.

4. An established clinical biomarker may appear correlated with SPINC1 and again with SPINC3 (as shown in GWAS plots where MAPT and SCAPER reappear). While it is stated that VAEs can discover orthogonal latent features, is this true in practice? A convincing experiment would be to show the specificity of hits between the latent features and more generally to look at the polygenic architecture (by estimating genetic correlations). And if this is also true or not true for EDFs.

We thank the reviewer for this question. As suggested by the reviewer, we computed the genetic correlations between 5 SPINCs as well as the genetic correlation between 5 EDFs (Table 4 in response and Supplementary Table 33 in paper). We observed that the genetic correlations between 5 SPINCs are much smaller than the genetic correlations between 5 EDFs (mean absolute values: 0.15 vs 0.57 and median absolute values: 0.15 vs 0.63). Furthermore, the maximum genetic correlation between SPINCs is 0.3177 (0.0494), which is between 5th and 3rd SPINCs, while EDFs max genetic correlation is 0.8962 (0.0535). In the case of PLENCs, we observed that the genetic correlations between 5 PLENCs are much smaller than the genetic correlations between 5 EDFs (mean absolute values: 0.15 vs 0.52 and median absolute values: 0.25 vs 0.59). Furthermore, the maximum genetic correlation between PLENCs is -0.5335 (0.0958), which is between 3rd and 2nd PLENCs, while the maximum genetic correlation between EDFs is close to 1 ($r_g=1.0031$ (0.0854)). It is worth mentioning that REGLE embedding tries to distangle the latent features while minimizing the reconstruction error, thus, one not always have distangle embeddings; however, in the case of SPINCs and PLENCs we observed that embedding have a very low phenotypic correlation and genetic correlation between embedding is much smaller than EDFs. Furthermore, the genetic correlation can be affected by the set of covariates that is considered. For example, if two phenotypes are genetically uncorrelated but when adjusted with a heritable covariate then these two phenotypes' genetic correlation under the heritable covariates is not zero anymore.

Pheno1	Pheno2	Genetic Corr (r_g)
SPINC ₂	SPINC ₁	0.1559 (0.0407)

SPINC ₃	SPINC ₁	-0.3087 (0.0308)
SPINC ₃	SPINC ₂	0.0364 (0.0264)
SPINC ₄	SPINC ₁	0.234 (0.0356)
SPINC ₄	SPINC ₂	0.0061 (0.0412)
SPINC ₄	SPINC ₃	0.1495 (0.0334)
SPINC ₅	SPINC ₁	-0.1951 (0.0381)
SPINC ₅	SPINC ₂	0.0811 (0.0333)
SPINC ₅	SPINC ₃	0.3177 (0.0494)
SPINC ₅	SPINC ₄	0.0306 (0.0444)
FEV ₁	FEF25-75%	0.7521 (0.0439)
FVC	FEF25-75%	0.369 (0.0286)
FVC	FEV ₁	0.8962 (0.0535)
PEF	FEF25-75%	0.699 (0.0444)
PEF	FEV ₁	0.6698 (0.0451)
PEF	FVC	0.4112 (0.0349)
FEV ₁ /FVC	FEF25-75%	0.8174 (0.0536)
FEV ₁ /FVC	FEV ₁	0.4279 (0.0297)
FEV ₁ /FVC	FVC	-0.0613 (0.0256)
FEV ₁ /FVC	PEF	0.5925 (0.036)
Peak to peak time	Absence of notch	-0.5783 (0.1515)
Position of notch	Absence of notch	-0.0413 (0.0682)
Position of notch	Peak to peak time	0.0389 (0.0972)
Position of peak	Absence of notch	0.7017 (0.0809)
Position of peak	Peak to peak time	-0.9171 (0.1458)
Position of peak	Position of notch	0.0861 (0.0661)
Position of shoulder	Absence of notch	0.6078 (0.0851)
Position of shoulder	Peak to peak time	-0.9501 (0.1391)
Position of shoulder	Position of notch	0.2833 (0.0653)
Position of shoulder	Position of peak	1.0031 (0.0854)
PLENC ₂	PLENC ₁	0.2917 (0.0689)

PLENC ₃	PLENC ₁	-0.0781 (0.0541)
PLENC ₃	PLENC ₂	-0.5335 (0.0958)

PLENC ₄	PLENC ₁	0.2868 (0.0874)
PLENC ₄	PLENC ₂	-0.1771 (0.1072)
PLENC ₄	PLENC ₃	0.2404 (0.094)
PLENC ₅	PLENC ₁	0.2755 (0.0687)
PLENC ₅	PLENC ₂	0.2941 (0.0815)
PLENC ₅	PLENC ₃	-0.0207 (0.0925)
PLENC ₅	PLENC ₄	0.0621 (0.1108)

Table 4. Genetic correlation between SPINCs, PLENCs and EDFs obtained by LDSC. We update the discussion (page 19) as follow:

“As a result, the PRSs of the learned encoding have lower correlation and contain relatively orthogonal genetic signals compared to EDFs (Supplementary Table 33).”

- The comparisons of the GWAS hits SPINCs to Shrine et al. and the NHGRI GWAS catalog is interesting but there is a concern that the Shrine et al. study overlaps UKBB as would the GWAS catalog (unless the authors are excluding this).

We performed a similar analysis to Table 1 where we used SpiroMeta, which excluded UK Biobank, instead of the Shrine et al. study (Supplementary Table 14 in manuscript and Table 5 in response to the reviewer). We observed similar results.

Method (# traits)	Sample size	Total	Known (%)	Novel (%)
GWAS Catalog + SpiroMeta	-	1237	-	-
SpiroMeta	-	29	-	-
Spirogram EDFs (5)	325K	613	529 (86%)	84 (14%)
Spirogram PCA (5)	325K	412	380 (92%)	32 (8%)
Spirogram cubic spline (5)	325K	435	384 (88%)	51 (12%)
SPINCs (5)	325K	575	469 (81%)	106 (18%)
EDFs+RSPINCs (7)	325K	659	540 (82%)	119 (18%)

Table 5. Comparison of GWAS significant loci. For lung function and spirometers, expert-defined features (EDFs) are FEV₁, FVC, FEV₁/FVC, PEF, and FEF_{25-75%}, and “known” and “novel” is in reference to lung function loci in GWAS Catalog and SpiroMeta.

We updated the Result section (page 12) and added Supplementary Table 14:

"Furthermore, we observed similar results when we used nonlinear cubic spline coefficients instead of linear PCs as a baseline (Supplementary Table 13) or when we excluded UK Biobank samples from Shrine et al. [25] (Supplementary Table 14)."

6. How correlated are the learned representations with each other (eg. do signals re-appear in more than one latent factor)?

The learned representations have extremely small phenotypic correlation for REGLE (Supplementary Figures 5-6 for spiromgrams and PPG, but this may not be true for other modalities) and PC. However, as stated above, even if two phenotypes are uncorrelated in phenotypic values, they still can have non-zero genetic correlations. Thus, REGLE learned representations are uncorrelated but have shared genetic signals and some signals may reappear between latent factors. We made this point clear in the Discussion and Methods (p xx):

"Although the phenotypic correlation between embeddings for REGLE and PCA is extremely low (Supplementary Figure 5), the genetic correlation is not necessarily zero (Supplementary Figure 33). It is worth noting that REGLE embeddings have lower genetic correlation and contain relatively orthogonal genetic signals compared to EDFs."

7. It is discussed that there are no guarantees in the direction (sign) of the learned latent features. While it is true we can re-orient eg. GWAS effects for relatively known phenotypes, would this not be a source of ambiguity for novel phenotypes?

We consider two possible scenarios: First, we want to utilize the novel phenotypes from REGLE for clinical applications. Second, we want to utilize the phenotypes for downstream genetic analyses such as PRS. In the case of the latter, we can utilize the REGLE result as-is. However, in the case of the former, we need to make sure that we use the same phenotypes and thus we need to align the novel phenotypes from different datasets or different runs of the model. We could use anchor phenotypes relevant to the organ system under study (e.g. asthma or COPD for spiromgrams) to compute the correlation between embeddings and anchor phenotypes, and then reorder the phenotypes to obtain the same correlation order. Alternatively, we could directly compute the pairwise phenotypic correlations between two different datasets or runs and align based on the maximum correlation (as is performed in Supplementary Figure 3).

Finally, it is worth mentioning that this issue occurs in the case of PCA as well as REGLE. Principal component analysis algorithms are deterministic, but the solutions are not unique. For example, you could easily change the sign of an eigenvector without altering the PCA.

Minor comments:

1. What is the scale of the reconstruction error? (e. g. vol^2)

As we are reconstructing both flow-time and volume-time, the reconstruction is a weighted scale of flow-time reconstruction error which has vol^2/s^2 and volume-time reconstruction error which has a vol^2 scale. Thus, the reconstruction error will not have one scale.

2. The VAE learns both the mean and variance. Is the variance information used at all in this study?

In this version of the work, we did not utilize the variance. We could utilize variance to provide uncertainty on the VAE embeddings or perform sampling using mean and variance. However, sampling the latent embeddings will introduce stochasticity which we decide to avoid.

We made this clear in the Methods section (p 26):

“It is worth mentioning that the learned variance for VAE is not utilized.”

Reviewer #3

Remarks to the Author:

The authors present a method to improve phenotyping in genome-wide association studies that uses unsupervised deep representation learning and show that this method reveals structure in multidimensional data that is missed by existing expert-defined features. They show that this uncaptured structure implicates new associated genetic loci that have been missed in previous genetic association studies of lung and cardiovascular data. This is an important contribution to the field and paves the way for more in-depth phenotyping leading to greater capture of genetic effects for traits investigated by multidimensional data.

The method is very clearly described and the results clearly demonstrate the advantage of the method to extract information from high-dimensional clinical data in the two scenarios demonstrated in order to find new genetic loci and improve the accuracy of risk scores. The interpretation of RSPINCs is also interesting on how different RSPINCs affect different parts of the blow curve.

I have a few queries:

The use of VAE seems unnecessarily complex, couldn't the spiograms/PPGs be adequately modelled by a simple function with a similar number of parameters to a SPINC/PLENC (the examples in refs. 1-5 are more complex HDCC where deep learning would seem more necessary). It's not clear to me how the introduction of stochasticity and forcing of disentanglement helps modelling the relatively simple form of spiograms/PPGs. The fact that the spiograms can be reconstructed with just 5 SPINCs makes me wonder if they could equally be accurately modelled with a cubic spline with 5 degrees of freedom?

We thank the reviewer for raising this question. As suggested by the reviewer, we fit cubic splines with one knot to generate exactly 5 spline coefficients, as implemented in SciPy using the FITPACK library. We observed that the GWAS on SPINCs coordinates generated more significant loci than the GWAS on spline coefficients, while the GWAS on spline coefficients generated more significant loci than GWAS on linear PCs.

We added the following sentence to Results section 2.7 and added Supplementary Table 13:

"Furthermore, we observed similar results when we used nonlinear cubic spline coefficients instead of linear PCs as a baseline (Supplementary Table 13) or when we excluded UK Biobank samples from Shrine et al. [25] (Supplementary Table 14)."

Furthermore, we compared the PRS result of cubic spline fitting for asthma and COPD and observed a similar pattern. For both asthma and COPD, SPINCs and EDFs+RSPINCs outperformed the cubic spline coefficients in all metrics we considered, while the cubic spline coefficients outperformed the PCA baseline.

We added the following sentence in Results and added new rows for cubic spline coefficients PRS in Supplementary Tables 18 and 19:

"Finally, we observed that for both asthma and COPD, both SPINCs and EDFs+RSPINCs PRS outperforms the PRS generated by baseline dimensionality reduction methods such as linear PCA or nonlinear cubic spline fitting (Supplementary Tables 18 and 19)."

We also added a detailed description of cubic spline fitting in Methods in the "Principal components and cubic spline coefficients" section:

"As baseline methods for dimensionality reduction, we performed principal component analysis (PCA) and cubic spline fitting on spirometry. For PCA we concatenated volume-time and flow-time curves and used both as inputs, while for cubic spline fitting we used only volume-time curves as cubic splines perform better for "smoother" curves. To match the number of EDFs and the dimension of SPINCs, we generated 5 principal components (PCs) and 5 cubic spline coefficients. We used 1 knot and cubic curves for spline fitting to generate exactly 5 coefficients, where the knot position was chosen at the 20% position to better capture the complexity at the beginning of the volume-time curves. Python scikit-learn package was used for PCA and SciPy was used for spline fitting."

Finally, it is worth mentioning that REGLE and PCA are general methods that can be applied to HDCD of any dimension, while cubic spline fitting is only applicable to relatively simple 1-dimensional HDCD in the curve form. Why 80/20 training/validation sets?

80%-20% split is a typical train-validation split used in modern machine learning, which was also used in our previous work (Cosentino et al. 2023 Nature Genetics). It is worth mentioning that this is done only for individuals with valid spirometry in the context of training SPINCs or RSPINCs models. Please note that for PRS performance evaluation in UK Biobank on asthma and COPD, we used completely separate test individuals (without valid spirometry) for fair comparison, ensuring that the PRS performance results are not biased (Supplementary Figure 1). We used a similar procedure for PPG as well (Supplementary Figure 4).

We updated the Methods section of manuscript as follows:

“We then subdivide all European ancestry individuals processed this way into a 80% training set and a 20% validation set similar to Cosentino et al. [6].”

Why is there no investigation of EDF + RPLENCs for PPGs?

We thank the reviewer for raising this point. We added the EDF+RPLENCs for PPG for GWAS loci and PRS. We updated Table 1 to include the EDF+RPLENCs detected and novel loci. We added PRS results to Supplementary Tables 23-24.

We added the following sentences in the "Overview of REGLE on PPGs" section in Results for the new EDFs+RPLENCs experiments:

"Similarly to spiograms, we also constructed residual photoplethysmogram encodings (RPLENCs) by injecting 5 PPG EDFs (the absence of notch, the position of notch, the position of peak, the position of shoulder, and the peak-to-peak time)."

We have added Supplementary Figure 10 (RPLENCs Kaplan-Meier curves), Supplementary Figures 25-26 (RPLENCs Manhattan plots), Supplementary Figures 40-41 (RPLENCs GARFIELD), and Supplementary Tables 31-32 (RPLENCs PheWAS) for the new RPLENCs experiments. We also updated Supplementary Table 1 (hyperparameters), Supplementary Table 3 (reconstruction errors), Supplementary Tables 7-8 (phenotype correlation), Supplementary Table 9 (survival analysis), and Supplementary Table 10 (S-LDSC) to include the RPLENCs results.

We have added the following sentences in the "High association between REGLE encodings and UKB phenotypes PRSs" section in Results for PheWAS on RPLENCs coordinates:

"For (R)PLENCs, we observed all coordinates show significant correlation with different traits including blood traits (red blood cell, eosinophil count, age high blood pressure diagnosed, and hemoglobin concentration), PPG traits (pulse rate and pulse wave reflection index), ECG traits (QRS duration and P duration), blood pressure, and vascular/heart problems (Supplementary Tables 29 to 32). The strongest correlation was obtained from 1st PLENCs coordinate with pulse rate ($R=-0.67$; $P\leq 1.00E-300$) and ECG heart rate ($R=-0.52$; $P\leq 1.00E-300$) and 1st RPLENCs coordinate with Pulse wave peak to peak time ($R=-0.31$; $P\leq 1.00E-300$)."

We have also added RPLENCs architecture details in Supplementary Notes and added the following sentences in Methods:

"RPLENCs are generated similarly to RSPINCs where we inject 5 EDFs directly to the sampled output of the bottleneck layer. See Supplementary Figure 44 and "PLENCs model architecture" and "RPLENCs model architecture" in Supplementary Notes for full details."

Line 247-248 - how does the hazard per SD of SPINC/PLENC compare to hazard per SD FEV1 or FVC?

SPINC₃ has a hazard ratio of 0.68 which is similar to FVC hazard ratio of 0.68 and higher (worse) than FEV₁ with hazard ratio of 0.64, implying the hazard of death decreased by 32% (FVC and SPINC₃) and 36% (FEV₁) per one standard deviation increase of each phenotypic values. While in the case of PLENCs, PLENC₂ has a hazard ratio of 0.76 compared to "Position of Shoulder" has a hazard ratio of 1.14 which implies the hazard of death decreased by 24% per one standard deviation increase of PLENC₂ values while in the case of "Position of Shoulder", death increased by 14% per one standard deviation increase. All values are reported in the Supplementary Table 9 (Table 6 in response to the reviewer). SPINC_s/PLENC_s are also strongly associated with mortality; the PLENCs are superior to the EDFs.

Risk	Hazard Ratio	Lower 95 % CI	Upper 95 % CI	P
SPINC ₁	1.08565	1.0516	1.1208	4.16E-07
SPINC ₂	1.13457	1.1010	1.1692	1.74E-16
SPINC ₃	0.67943	0.6534	0.7065	1.57E-83
SPINC ₄	1.07763	1.0449	1.1114	2.00E-06
SPINC ₅	1.00962	0.9794	1.0408	5.37E-01
RSPINC ₁	0.98222	0.9527	1.0127	2.49E-01
RSPINC ₂	1.06364	1.0316	1.0967	7.81E-05
FEV ₁	0.63971	0.6148	0.6656	8.10E-108
FVC	0.68493	0.6558	0.7153	2.15E-65
PEF	0.75200	0.7275	0.7773	7.72E-64
FEV ₁ /FVC	0.80562	0.7859	0.8258	1.07E-65
FEF25-75%	0.68917	0.6630	0.7164	2.23E-79

PLENC ₁	0.87865	0.8339	0.9258	1.24E-06
PLENC ₂	0.76070	0.7208	0.8029	2.83E-23
PLENC ₃	1.07235	1.0177	1.1299	8.82E-03
PLENC ₄	1.01382	0.9615	1.0690	0.611623
PLENC ₅	0.99256	0.9419	1.0459	0.779784
Absence of Notch	1.07081	1.0245	1.1192	2.43E-03

Position of Notch	1.12913	1.0766	1.1842	5.89E-07
Position of Peak	1.11444	1.0536	1.1788	1.54E-04
Position of Shoulder	1.14546	1.0847	1.2096	1.04E-06
Peak to Peak Time	0.96973	0.9202	1.0219	2.50E-01

Table 6. Survival analysis hazard ratios per 1 standard deviation for SPINC, RSPINC, PLENC, and EDF risk scores.

We updated the manuscript as follow (p 10):

“SPINC hazard ratios are similar to FVC and slightly worse than FEV₁ (0.64 vs 0.68; Supplementary Table 9), given that FEV₁ is a known clinical feature for COPD, REGLE can extract important clinical features from HDCD.”

Line 262: How does heritability of (R)SPINC/PLENCs compare to heritability of EDFs?

For spirometers, the maximum heritability of EDFs and SPINC are not significantly different, while for PPG the maximum heritability of PLENCs is significantly higher than EDFs (0.1288 (0.0087) vs 0.0762 (0.0074)). As expected, the heritabilities of EDFs are very similar to each other as they are highly correlated.

Pheno	h ² _g
SPINC ₁	0.1302 (0.0071)
SPINC ₂	0.2481 (0.0132)
SPINC ₃	0.1604 (0.0066)
SPINC ₄	0.0746 (0.0055)
SPINC ₅	0.0428 (0.0029)
FEV ₁	0.1884 (0.0079)
FVC	0.1961 (0.0074)
PEF	0.1222 (0.0063)
FEV ₁ /FVC	0.2026 (0.0101)

FEF25%-75%	0.2445 (0.0115)
PLENC ₁	0.1288 (0.0087)

PLENC ₂	0.0682 (0.0072)
PLENC ₃	0.0604 (0.0067)
PLENC ₄	0.0366 (0.0067)
PLENC ₅	0.0704 (0.0074)
Absence of Notch	0.0409 (0.0061)
Position of Notch	0.0762 (0.0074)
Position of Peak	0.0662 (0.0068)
Position of Shoulder	0.0699 (0.0073)
Peak to Peak Time	0.024 (0.0066)

Table 7. S-LDSC results on SPINCs, RSPINCs, PLENCs, and EDFs GWAS. We computed the S-LDSC SNP-heritability. Values in parentheses are the standard error of the mean (s.e.m) obtained from S-LDSC.

Line 272-273: Shrine et al. used 5E-9, you are using 5E-8, shouldn't you compare number of loci found and same level of significance?

All results reported in Table 1 of the manuscript (except for GWAS Catalog search results) used the exact same P-value threshold of 5E-8. We downloaded the full GWAS summary statistics from [Shrine et al. Nature Genetics 2023] and applied the exact same P-value threshold and merged hits using the exact same criteria as described in the manuscript, to compute the number of independent genome-wide significant loci for fair comparison. Please also note that [Shrine et al. Nature Genetics 2023] uses a much larger dataset (581K individuals) compared to our GWAS in UK Biobank (325K individuals). We updated the following sentence in the "Genome-wide association studies and polygenic risk scores" section in Methods for clarification:

"To determine the known lung function loci from previous literature, we extracted all significant loci from Shrine et al. by downloading the full GWAS summary statistics and merging hits using the exact same criteria and P-value threshold described above, and searched for lung function-related keywords in the NHGRI-EBI GWAS Catalog."

I don't see how the GARFIELD analysis demonstrates any improvement over EDFs?

We performed GARFIELD analysis to investigate the functional analysis of REGLE embeddings. This analysis aids us in understanding the tissue and cell specific signals. More specifically, we aim to make sure the novel findings by REGLE are enriched in expected cell-types and tissues. In fact, for GARFIELD analysis, we never compared REGLE results with EDFs.

I don't see what the causality analysis adds, aren't COPD and hypertension just defined by thresholds of spiromograms and PPGs respectively?

The main point of causality analysis is to further validate that the latent factors/embeddings capture the true signal and not some confounding signal of spiromograms and PPG. Given that this is not the main analysis of the paper, we have removed the causality analysis from the current version of the manuscript.

Line 72-74: 2 and 3 are the same issue, redundant multiple testing

We combined the 2 and 3 to one as a redundant multiple hypothesis testing issue.

Line 256-257: could just say Supplementary Figures 12-23

We changed the text as suggested. Thank you.

Between Supplementary Table 12 & 13 the captions have switched from being above the table to below it.

We fixed these issues.

Would Supplementary Tables 14-17 suggest it is better to use SPINCs than EDF+RSPINCs?

It is correct that SPINCs has better PRS and GWAS (higher statistical power) compared to EDF+RSPINCs (same holds for PPG), thus, if the main goal is to have the best GWAS or PRS, it is better to utilize SPINCs/PLENCs. However, a large fraction of the time, clinicians and researchers have set of known features (i.e., EDFs) that are considered important for them and embeddings such as RSPINCs and RPLENCs provide additional signals from HDCD to further improve their downstream tasks. We consider following main advantages of EDF+RSPINCs (same hold for EDF+RPLENCs):

- 1) EDF+RSPINCs is more interpretable and easier to use in clinical settings as these embeddings utilize known EDFs.**

- 2) EDF+RSPINCs provides additional information in HDCD that previously was not utilized via EDF.
- 3) EDF+RSPINCs outperforms EDFs in both PRS and GWAS statistical power while being slightly worse than SPINCs.

In short, EDF+RSPINCs combine the benefits of EDF and SPINCs in one model. We updated Discussion to make this point clear (page 19):

“Although these models have slightly lower GWAS power and PRS performance compared to non-residual REGLE (i.e. SPINCs vs RSPINCs; Table 1 and Supplementary Tables 14, 18, 19, 21, 23 and 24), the residual models combine the benefits of EDFs and REGLE into one model and thus provide a mechanism to build upon and improve existing clinical practices with the extra information provided from the residual features.”

Decision Letter, first revision:

23rd Feb 2024

Dear Farhad,

Thank you for submitting your revised manuscript "Unsupervised representation learning improves genomic discovery for lung function and respiratory disease prediction" (NG-A62621R2). It has now been seen by the original referees and their comments are below. The reviewers find that the paper has improved in revision, and therefore we'll be happy in principle to publish it in Nature Genetics, pending minor revisions to satisfy the referees' final requests and to comply with our editorial and formatting guidelines.

Sincerely,

Michael Fletcher, PhD
Senior Editor, Nature Genetics

ORCID: 0000-0003-1589-7087

Reviewer #1 (Remarks to the Author):

The authors have addressed all of my questions and concerns. I feel that the pieces that I did not quite follow have been clarified, and the description of this work in relationship with prior work is also clearly described.

Reviewer #2 (Remarks to the Author):

The authors proposed an approach to use an unsupervised deep learning model to learn latent factors of higher dimensional health data modalities which has the potential of leading to the discovery of novel associated genomic variants. The reviewers raised comments about the rationale behind the choice of the number of latent factors, the characteristics of those latent factors given that they are learned in an unsupervised setting, and possible interpretations of the downstream analysis.

A point raised by more than one reviewer on whether the neural network introduces correlations among the latent factors previously absent in EDFs, leading to concerns about the downstream analysis, was sufficiently addressed by the genetic correlation result. A related point was if the VAE latent factors were more interesting than the same number of PCA factors given the reconstruction error appeared visually similar. The authors have demonstrated in several ways now (motivations that EDFs themselves are non-linear, more hits including those from PCA, chi square statistics, PRS, baseline of fitting splines) that the latent factors are more interesting than those obtained by prior approaches.

From my perspective, the reinforcing results sufficiently explore all the points raised in the review and believe that readers will benefit from the publication of this research.

Reviewer #3 (Remarks to the Author):

The authors have addressed all my original criticisms adequately and have convinced me that the method is an improvement over existing techniques.

Final Decision Letter:

13th Jun 2024

Dear Farhad,

I am delighted to say that your manuscript "Unsupervised representation learning on high-dimensional clinical data improves genomic discovery and prediction" has been accepted for publication in an upcoming issue of Nature Genetics.

Your paper will be published online after we receive your corrections and will appear in print in the next available issue. You can find out your date of online publication by contacting the Nature Press Office (press@nature.com) after sending your e-proof corrections.

Please note that *Nature Genetics* is a Transformative Journal (TJ). Authors may publish their research with us through the traditional subscription access route or make their paper immediately open access through payment of an article-processing charge (APC). Authors will not be required to make a final decision about access to their article until it has been accepted. Find out more about Transformative Journals

Authors may need to take specific actions to achieve compliance with funder and institutional open access mandates. If your research is supported by a funder that requires immediate open access (e.g. according to Plan S principles) then you should select the gold OA route, and we will direct you to the compliant route where possible. For authors selecting the subscription publication route, the journal's standard licensing terms will need to be accepted, including [a href="https://www.nature.com/nature-portfolio/editorial-policies/self-archiving-and-license-to-publish"](https://www.nature.com/nature-portfolio/editorial-policies/self-archiving-and-license-to-publish). Those licensing terms will supersede any other terms that the author or any third party may

assert apply to any version of the manuscript.

If you have not already done so, we strongly recommend that you upload the step-by-step protocols used in this manuscript to protocols.io. protocols.io is an open online resource that allows researchers to share their detailed experimental know-how. All uploaded protocols are made freely available and are assigned DOIs for ease of citation. Protocols can be linked to any publications in which they are used and will be linked to from your article. You can also establish a dedicated workspace to collect all your lab Protocols. By uploading your Protocols to protocols.io, you are enabling researchers to more readily reproduce or adapt the methodology you use, as well as increasing the visibility of your protocols and papers. Upload your Protocols at <https://protocols.io>. Further information can be found at <https://www.protocols.io/help/publish-articles>.

Sincerely,

Michael Fletcher, PhD
Senior Editor, Nature Genetics
ORCID: 0000-0003-1589-7087